# FAST DIFFERENTIABLE MATRIX SQUARE ROOT

**Yue Song, Nicu Sebe & Wei Wang**
Department of Information Engineering and Computer Science (DISI)
University of Trento
Trento, TN 38123, Italy
{yue.song}@unitn.it

## ABSTRACT

Computing the matrix square root or its inverse in a differentiable manner is important in a variety of computer vision tasks. Previous methods either adopt the Singular Value Decomposition (SVD) to explicitly factorize the matrix or use the Newton-Schulz iteration (NS iteration) to derive the approximate solution. However, both methods are not computationally efficient enough in either the forward pass or in the backward pass. In this paper, we propose two more efficient variants to compute the differentiable matrix square root. For the forward propagation, one method is to use Matrix Taylor Polynomial (MTP), and the other method is to use Matrix Padé Approximants (MPA). The backward gradient is computed by iteratively solving the continuous-time Lyapunov equation using the matrix sign function. Both methods yield considerable speed-up compared with the SVD or the Newton-Schulz iteration. Experimental results on the de-correlated batch normalization and second-order vision transformer demonstrate that our methods can also achieve competitive and even slightly better performances. The code is available at https://github.com/KingJamesSong/FastDifferentiableMatSqrt.

## 1 INTRODUCTION

Consider a positive semi-definite matrix $\boldsymbol{A}$. The principle square root $\boldsymbol{A}^{\frac{1}{2}}$ and the inverse square root $\boldsymbol{A}^{-\frac{1}{2}}$ (often derived by calculating the inverse of $\boldsymbol{A}^{\frac{1}{2}}$) are mathematically of practical interests, mainly because some desired spectral properties can be obtained by such transformations. An exemplary illustration is given in Fig. 1 (a). As can be seen, the matrix square root can shrink/stretch the feature variances along with the direction of principle components, which is known as an effective spectral normalization for covariance matrices. The inverse square root, on the other hand, can be used to whiten the data, *i.e.,* make the data has a unit variance in each dimension. Due to the appealing spectral properties, computing the matrix square root or its inverse in a differentiable manner arises in a wide range of computer vision applications, including covariance pooling (Lin & Maji, 2017; Li et al., 2018; Song et al., 2021), decorrelated batch normalization (Huang et al., 2018; 2019; 2020), and Whitening and Coloring Transform (WCT) (Li et al., 2017b; Cho et al., 2019; Choi et al., 2021).

To compute the matrix square root, the standard method is via Singular Value Decomposition (SVD). Given the real Hermitian matrix $\boldsymbol{A}$, its matrix square root is computed as:

$$\boldsymbol{A}^{\frac{1}{2}} = (\boldsymbol{U}\boldsymbol{\Lambda}\boldsymbol{U}^T)^{\frac{1}{2}} = \boldsymbol{U}\boldsymbol{\Lambda}^{\frac{1}{2}}\boldsymbol{U}^T \tag{1}$$

where $\boldsymbol{U}$ is the eigenvector matrix, and $\boldsymbol{\Lambda}$ is the diagonal eigenvalue matrix. As derived by Ionescu et al. (2015b), the partial derivative of the eigendecomposition is calculated as:

$$\frac{\partial l}{\partial \boldsymbol{A}} = \boldsymbol{U}\Big(\boldsymbol{K}^T \odot (\boldsymbol{U}^T \frac{\partial l}{\partial \boldsymbol{U}}) + (\frac{\partial l}{\partial \boldsymbol{\Lambda}})_{\text{diag}}\Big)\boldsymbol{U}^T \tag{2}$$

where $l$ is the loss function, $\odot$ denotes the element-wise product, and $()_{\text{diag}}$ represents the operation of setting the off-diagonal entries to zero. Despite the long-studied theories and well-developed algorithms of SVD, there exist two obstacles when integrated into deep learning frameworks. One issue is the back-propagation instability. For the matrix $\boldsymbol{K}$ defined in Eq. (2), its off-diagonal entry is $K_{ij}=1/(\lambda_i-\lambda_j)$, where $\lambda_i$ and $\lambda_j$ are involved eigenvalues. When the two eigenvalues are close

and small, the gradient is very likely to explode, *i.e.,* $K_{ij} \to \infty$. This issue has been solved by some methods that use approximation techniques to estimate the gradients (Wang et al., 2019; 2021; Song et al., 2021). The other problem is the expensive time cost of the forward eigendecomposition. As the SVD is not supported well by GPUs, performing the eigendecomposition on the deep learning platforms is rather time-consuming. Incorporating the SVD with deep models could add extra burdens to the training process. Particularly for batched matrices, modern deep learning frameworks, such as Tensorflow and Pytorch, give limited optimization for the matrix decomposition within the mini-batch. A (parallel) for-loop is inevitable for conducting the SVD one matrix by another. However, how to efficiently perform the SVD in the context of deep learning has not been touched.

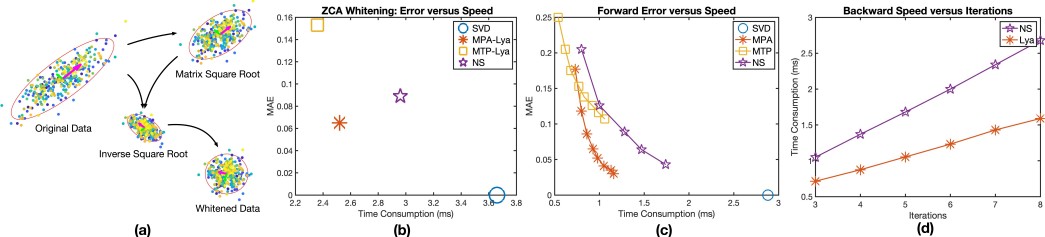

Figure 1: **(a)** Exemplary visualization of the matrix square root and its inverse. Given the original data $\boldsymbol{X} \in \mathbb{R}^{2 \times n}$, the matrix square root stretches the data along the axis of small variances and squeezes the data in the direction with large variances, serving as a spectral normalization for covariance matrices. The inverse square root, on the other hand, can be used to transform the data into the uncorrelated structure, *i.e.,* have the unit variance in each dimension. **(b)** The comparison of error and speed using the setting of our ZCA whitening experiment. Our MTP and MPA are faster than the SVD and the NS iteration, and our MPA is more accurate than the NS iteration. **(c)** The comparison of speed and error in the forward pass (FP). The iteration times of the NS iteration range from 3 to 7, while the degrees of our MTP and MPA vary from 6 to 18. Our MPA computes the more accurate and faster matrix square root than the NS iteration, and our MTP enjoys the fastest calculation speed. **(d)** The speed comparison in the backward pass (BP). Our Lyapunov solver is more efficient than the NS iteration as fewer matrix multiplications are involved.

To avoid explicit eigendecomposition, one commonly used alternative is the Newton-Schulz iteration (NS iteration) (Schulz, 1933; Higham, 2008). It modifies the ordinary Newton iteration by replacing the matrix inverse but preserves the quadratic convergence. Compared with SVD, the NS iteration is rich in matrix multiplication and more GPU-friendly. Thus, this technique has been widely used to approximate the matrix square root in different applications (Lin & Maji, 2017; Li et al., 2018; Huang et al., 2019). The forward computation relies on the following coupled iterations:

$$\boldsymbol{Y}_{k+1} = \frac{1}{2}\boldsymbol{Y}_k(3\boldsymbol{I} - \boldsymbol{Z}_k\boldsymbol{Y}_k), \boldsymbol{Z}_{k+1} = \frac{1}{2}(3\boldsymbol{I} - \boldsymbol{Z}_k\boldsymbol{Y}_k)\boldsymbol{Z}_k \tag{3}$$

where $\boldsymbol{Y}_k$ and $\boldsymbol{Z}_k$ converge to the matrix square root $\boldsymbol{A}^{\frac{1}{2}}$ and the inverse square root $\boldsymbol{A}^{-\frac{1}{2}}$, respectively. Since the NS iteration only converges locally, we need to pre-normalize the initial matrix and post-compensate the resultant approximation as:

$$\boldsymbol{Y}_0 = \frac{1}{||\boldsymbol{A}||_{\mathrm{F}}}\boldsymbol{A}, \ \boldsymbol{A}^{\frac{1}{2}} = \sqrt{||\boldsymbol{A}||_{\mathrm{F}}}\boldsymbol{Y}_k. \tag{4}$$

Each forward iteration involves 3 matrix multiplications, which is more efficient than the forward pass of SVD. The backward pass of the NS iteration is calculated as:

$$\frac{\partial l}{\partial \boldsymbol{Y}_k} = \frac{1}{2}\Big(\frac{\partial l}{\partial \boldsymbol{Y}_{k+1}}(3\boldsymbol{I} - \boldsymbol{Y}_k\boldsymbol{Z}_k) - \boldsymbol{Z}_k\frac{\partial l}{\partial \boldsymbol{Z}_{k+1}}\boldsymbol{Z}_k - \boldsymbol{Z}_k\boldsymbol{Y}_k\frac{\partial l}{\partial \boldsymbol{Y}_{k+1}}\Big),$$

$$\frac{\partial l}{\partial \boldsymbol{Z}_k} = \frac{1}{2}\Big((3\boldsymbol{I} - \boldsymbol{Y}_k\boldsymbol{Z}_k)\frac{\partial l}{\partial \boldsymbol{Z}_k} - \boldsymbol{Y}_k\frac{\partial l}{\partial \boldsymbol{Y}_{k+1}}\boldsymbol{Y}_k - \frac{\partial l}{\partial \boldsymbol{Z}_{k+1}}\boldsymbol{Z}_k\boldsymbol{Y}_k\Big) \tag{5}$$

where the backward pass needs 10 matrix multiplications for each iteration. The backward gradients for the post-compensation and pre-normalization steps are computed as:

$$\frac{\partial l}{\partial \boldsymbol{A}}\Big|_{post} = \frac{1}{2||\boldsymbol{A}||_{\mathrm{F}}^{3/2}}\mathrm{tr}\Big(\big(\frac{\partial l}{\partial \boldsymbol{Y}_k}\big)^T\boldsymbol{Y}_k\Big)\boldsymbol{A},$$

$$\frac{\partial l}{\partial \boldsymbol{A}}\Big|_{pre} = -\frac{1}{||\boldsymbol{A}||_{\mathrm{F}}^3}\mathrm{tr}\Big(\big(\frac{\partial l}{\partial \boldsymbol{Y}_0}\big)^T\boldsymbol{A}\Big)\boldsymbol{A} + \frac{1}{||\boldsymbol{A}||_{\mathrm{F}}}\frac{\partial l}{\partial \boldsymbol{Y}_0} + \frac{\partial l}{\partial \boldsymbol{A}}\Big|_{post}. \tag{6}$$

These two steps take 4 matrix multiplications in total. Consider the fact that the NS iteration often takes 5 iterations to achieve reasonable performances (Li et al., 2018; Huang et al., 2019). The backward pass is much more time-costing than the backward algorithm of SVD. As seen from Fig. 1 (b) and (c), although the NS iteration outperforms SVD by 123% in the speed of forward pass, its overall time consumption only improves that of SVD by 17%. The speed improvement could be larger if a more efficient backward algorithm is developed.

To address the drawbacks of SVD and NS iteration, *i.e.* the low efficiency in either the forward or backward pass, we derive two methods **that are efficient in both forward and backward propagation** to compute the differentiable matrix square root. In the forward pass, we propose to use Matrix Taylor Polynomial (MTP) and Matrix Padé Approximants (MPA) to approximate the matrix square root. The former approach is slightly faster but the latter is more numerically accurate (see Fig. 1 (c)). Both methods yield considerable speed-up compared with the SVD or the NS iteration in the forward computation. For the backward pass, we consider the gradient function as a Lyapunov equation and propose an iterative solution using the matrix sign function. The backward pass costs fewer matrix multiplications and is more computationally efficient than the NS iteration (see Fig. 1 (d)). Through a series of numerical tests, we show that the proposed MTP-Lya and MPA-Lya deliver consistent speed improvement for different batch sizes, matrix dimensions, and some hyper-parameters (*e.g.,* degrees of power series to match and iteration times). Moreover, our proposed MPA-Lya consistently gives a better approximation of the matrix square root than the NS iteration. Besides the numerical tests, experiments on the decorrelated batch normalization and second-order vision transformer also demonstrate that our methods can achieve competitive and even better performances against the SVD and the NS iteration with the least amount of time overhead.

Finally, to promote the accessibility of our methods, the Pytorch implementation of all the utilized methods will be released upon acceptance.

## 2 DIFFERENTIABLE MATRIX SQUARE ROOT IN DEEP LEARNING

In this section, we first recap the previous approaches that compute the differentiable matrix square root and then discuss its usages in some applications of deep learning.

### 2.1 COMPUTATION METHODS

Ionescu et al. (2015b;a) first formulate the theory of matrix back-propagation, making it possible to integrate a spectral meta-layer into neural networks. Existing approaches that compute the differentiable matrix square root are mainly based on the SVD or NS iteration. The SVD calculates the accurate matrix square root but suffers from backward instability and expensive time cost, whereas the NS iteration computes the approximate solution but is more GPU-friendly. For the backward algorithm of SVD, several methods have been proposed to resolve this issue of gradient explosion (Wang et al., 2019; Dang et al., 2018; 2020; Wang et al., 2021; Song et al., 2021). Wang et al. (2019) propose to apply Power Iteration (PI) to approximate the SVD gradient. Recently, Song et al. (2021) propose to rely on Padé approximants to closely estimate the backward gradient of SVD.

To avoid explicit eigendecomposition, Lin & Maji (2017) propose to substitute SVD with the NS iteration. Following this work, Li et al. (2017a); Huang et al. (2018) adopt the NS iteration in the task of global covariance pooling and decorrelated batch normalization, respectively. For the backward pass of the differentiable matrix square root, Lin & Maji (2017) also suggest viewing the gradient function as a Lyapunov equation. However, their proposed exact solution is infeasible to compute practically, and the suggested Bartels-Steward algorithm (Bartels & Stewart, 1972) requires explicit eigendecomposition or Schur decomposition, which is again not GPU-friendly. By contrast, our proposed iterative solution using the matrix sign function is more computationally efficient and achieves comparable performances against the Bartels-Steward algorithm (see the Appendix).

### 2.2 APPLICATIONS

One successful application of the differentiable matrix square root is the Global Covariance Pooling (GCP), which is a meta-layer inserted before the FC layer of deep models to compute the matrix square root of the feature covariance. Equipped with the GCP meta-layers, existing deep mod-

els have achieved state-of-the-art performances on both generic and fine-grained visual recognition (Lin et al., 2015; Li et al., 2017a; Lin & Maji, 2017; Li et al., 2018; Wang et al., 2020a; Song et al., 2021). More recently, Xie et al. (2021) integrate the GCP meta-layer into the vision transformer (Dosovitskiy et al., 2020) to exploit the second-order statistics of the high-level visual tokens, which solves the issue that vision transformers need pre-training on ultra-large-scale datasets.

Another line of research proposes to use ZCA whitening, which applies the inverse square root of the covariance to whiten the feature, as an alternative scheme for the standard batch normalization (Ioffe & Szegedy, 2015). The whitening procedure, a.k.a decorrelated batch normalization, does not only standardize the feature but also eliminates the data correlation. The decorrelated batch normalization can improve both the optimization efficiency and generalization ability of deep neural networks (Huang et al., 2018; Siarohin et al., 2018; Huang et al., 2019; Pan et al., 2019; Huang et al., 2020; 2021; Ermolov et al., 2021).

The WCT (Li et al., 2017b) is also an active research field where the differentiable matrix square root and its inverse are widely used. In general, the WCT performs successively the whitening transform (using inverse square root) and the coloring transform (using matrix square root) on the multi-scale features to preserve the content of current image but carry the style of another image. During the past few years, the WCT methods have achieved remarkable progress in universal style transfer (Li et al., 2017b; Wang et al., 2020b), domain adaptation (Abramov et al., 2020; Choi et al., 2021), and image translation (Ulyanov et al., 2017; Cho et al., 2019).

Besides the three main applications discussed above, there are still some minor applications, such as semantic segmentation (Sun et al., 2021) and super resolution (Dai et al., 2019).

## 3 Fast Differentiable Matrix Square Root

This section presents the forward propagation and the backward computation of our differentiable matrix square root. For the inverse square root, it can be easily derived by computing the inverse of the matrix square root. Moreover, depending on the application, usually the operation of matrix inverse can be avoided by solving the linear system, *i.e.,* computing $A=B^{-1}C$ is equivalent to solving $BA=C$. Thanks to the use of LU factorization, such a solution is more numerically stable and usually much faster than the matrix inverse.

### 3.1 Forward Pass

For the forward pass, we derive two variants: the MTP and MPA. The former approach is slightly faster but the latter is more accurate. Now we illustrate these two methods in detail.

#### 3.1.1 Matrix Taylor Polynomial

We begin with motivating the Taylor series for the scalar case. Consider the following power series:

$$(1 - z)^{\frac{1}{2}} = 1 - \sum_{k=1}^{\infty} \left| \binom{\frac{1}{2}}{k} \right| z^k \tag{7}$$

where the notion $\binom{\frac{1}{2}}{k}$ denotes the binomial coefficients that involve fractions, and the series converges when $z<1$ according to the Cauchy root test. For the matrix case, the power series can be similarly defined by:

$$(I - Z)^{\frac{1}{2}} = I - \sum_{k=1}^{\infty} \left| \binom{\frac{1}{2}}{k} \right| Z^k \tag{8}$$

where $I$ is the identity matrix. Substituting $Z$ with $(I-A)$ leads to:

$$A^{\frac{1}{2}} = I - \sum_{k=1}^{\infty} \left| \binom{\frac{1}{2}}{k} \right| (I - A)^k \tag{9}$$

Similar with the scalar case, the power series converge only if $||(I - A)||_p<1$, where $|| \cdot ||_p$ denotes any vector-induced matrix norms. To circumvent this issue, we can first pre-normalize the matrix

Table 1: Comparison of forward operations.

| Op. | MTP | MPA | NS iteration |
|---|---|---|---|
| Mat. Mul. | $K-1$ | $(K-1)/2$ | $3 \times$ #iters |
| Mat. Inv. | 0 | 1 | 0 |

Table 2: Comparison of backward operations.

| Op. | Lya | NS iteration |
|---|---|---|
| Mat. Mul. | $6 \times$ #iters | $4 + 10 \times$ #iters |
| Mat. Inv. | 0 | 0 |

$A$ by dividing $||A||_{\mathrm{F}}$. This can guarantee the convergence as $||I - \frac{A}{||A||_{\mathrm{F}}}||_p < 1$ is always satisfied. Afterwards, the matrix square root $A^{\frac{1}{2}}$ is post-compensated by multiplying $\sqrt{||A||_{\mathrm{F}}}$. Integrated with these two operations, Eq. (9) can be re-formulated as:

$$A^{\frac{1}{2}} = \sqrt{||A||_{\mathrm{F}}} \cdot \left( I - \sum_{k=1}^{\infty} \left| \binom{\frac{1}{2}}{k} \right| (I - \frac{A}{||A||_{\mathrm{F}}})^k \right) \tag{10}$$

Truncating the series to a certain degree $K$ yields the MTP approximation for the matrix square root. For the MTP of degree $K$, $K-1$ matrix multiplications are needed.

### 3.1.2 MATRIX PADÉ APPROXIMANTS

The MTP enjoys the fast calculation, but it converges uniformly and sometimes suffers from the so-called "hump phenomenon", *i.e.,* the intermediate terms of the series grow quickly but cancel each other in the summation, which results in a large approximation error. Expanding the series to a higher degree does not solve this issue either. The MPA, which adopts two polynomials of smaller degrees to construct a rational approximation, is able to avoid this caveat. The coefficients of the MPA polynomials need to be pre-computed by matching to the corresponding Taylor series. Given the power series of scalar in Eq. (7), the coefficients of a $[M, N]$ scalar Padé approximant are computed by matching to the truncated series of degree $M+N+1$:

$$\frac{1 - \sum_{m=1}^{M} p_m z^m}{1 - \sum_{n=1}^{N} q_n z^n} = 1 - \sum_{k=1}^{M+N} \left| \binom{\frac{1}{2}}{k} \right| z^k \tag{11}$$

where $p_m$ and $q_n$ are the coefficients that also apply to the matrix case. This matching gives rise to a system of linear equations, and solving these equations directly determines the coefficients. The numerator polynomial and denominator polynomial of MPA are given by:

$$P_M = I - \sum_{m=1}^{M} p_m (I - \frac{A}{||A||_{\mathrm{F}}})^m, \ Q_N = I - \sum_{n=1}^{N} q_n (I - \frac{A}{||A||_{\mathrm{F}}})^n \tag{12}$$

Then the MPA for approximating the matrix square root is computed as:

$$A^{\frac{1}{2}} = \sqrt{||A||_{\mathrm{F}}} Q_N^{-1} P_M, \tag{13}$$

Compared with the MTP, the MPA trades off half of the matrix multiplications with one matrix inverse, which slightly increases the computational cost but converges more quickly and delivers better approximation abilities. Moreover, we note that the matrix inverse can be avoided, as Eq. (13) can be more efficiently and numerically stably computed by solving the linear system $Q_N A^{\frac{1}{2}} = \sqrt{||A||_{\mathrm{F}}} P_M$. According to Van Assche (2006), diagonal Padé approximants (*i.e.,* $P_M$ and $Q_N$ have the same degree) usually yield better approximation than the non-diagonal ones. Therefore, to match the MPA and MTP of the same degree, we set $M=N=\frac{K-1}{2}$.

Table 1 summarizes the forward computational complexity. As suggested in Li et al. (2018); Huang et al. (2019), the iteration times for NS iteration are often set as 5 such that reasonable performances can be achieved. That is, to consume the same complexity as the NS iteration does, our MTP and MPA can match to the power series up to degree 16. However, as depicted in Fig. 1 (c), our MPA achieves the better accuracy than NS iteration even at degree 8. This observation implies that our MPA is a better option in terms of both accuracy and speed than the NS iteration.

### 3.2 BACKWARD PASS

Though one can manually derive the gradient of the MPA and MTP, their backward algorithms are computationally expensive as they involve the matrix power up to degree $K$, where $K$ can be

arbitrarily large. Relying on the AutoGrad package of deep learning frameworks can be both time and memory-consuming. To attain a more efficient backward algorithm, we propose to iteratively solve the gradient equation using the matrix sign function. Given the matrix $\mathbf{A}$ and its square root $\boldsymbol{A}^{\frac{1}{2}}$, since we have $\boldsymbol{A}^{\frac{1}{2}}\boldsymbol{A}^{\frac{1}{2}}=\boldsymbol{A}$, a perturbation on $\boldsymbol{A}$ leads to:

$$\boldsymbol{A}^{\frac{1}{2}}d\boldsymbol{A}^{\frac{1}{2}} + d\boldsymbol{A}^{\frac{1}{2}}\boldsymbol{A}^{\frac{1}{2}} = d\boldsymbol{A} \tag{14}$$

Using the chain rule, the gradient function of the matrix square root satisfies:

$$\boldsymbol{A}^{\frac{1}{2}}\frac{\partial l}{\partial \boldsymbol{A}} + \frac{\partial l}{\partial \boldsymbol{A}}\boldsymbol{A}^{\frac{1}{2}} = \frac{\partial l}{\partial \boldsymbol{A}^{\frac{1}{2}}} \tag{15}$$

As pointed out in Lin & Maji (2017), Eq. (15) actually defines the continuous-time Lyapunov equation ($\boldsymbol{BX}+\boldsymbol{XB}=\boldsymbol{C}$) or a special case of Sylvester equation ($\boldsymbol{BX}+\boldsymbol{XD}=\boldsymbol{C}$). The closed-form solution is given by:

$$vec(\frac{\partial l}{\partial \boldsymbol{A}}) = \left(\boldsymbol{A}^{\frac{1}{2}} \otimes \boldsymbol{I} + \boldsymbol{I} \otimes \boldsymbol{A}^{\frac{1}{2}}\right)^{-1} vec(\frac{\partial l}{\partial \boldsymbol{A}^{\frac{1}{2}}}) \tag{16}$$

where $vec(\cdot)$ denotes unrolling a matrix to vectors, and $\otimes$ is the Kronecker product. Although the closed-form solution exists theoretically, it can not be computed in practice due to the huge memory consumption of the Kronecker product. Another approach to solve Eq. (15) is via Bartels-Stewart algorithm (Bartels & Stewart, 1972). However, it requires explicit eigendecomposition or Schulz decomposition, which is not GPU-friendly and computationally expensive.

We propose to use the matrix sign function and iteratively solve the Lyapunov equation. Solving the Sylvester equation via matrix sign function has been long studied in the literature of numerical analysis (Roberts, 1980; Kenney & Laub, 1995; Benner et al., 2006). One notable line of research is using the family of Newton iterations. Consider the following continuous Lyapunov function:

$$\boldsymbol{BX} + \boldsymbol{XB} = \boldsymbol{C} \tag{17}$$

where $\boldsymbol{B}$ refers to $\boldsymbol{A}^{\frac{1}{2}}$ in Eq. (15), $\boldsymbol{C}$ represents $\frac{\partial l}{\partial \boldsymbol{A}^{\frac{1}{2}}}$, and $\boldsymbol{X}$ denotes the seeking solution $\frac{\partial l}{\partial \boldsymbol{A}}$. Eq. (17) can be represented by the following block using a Jordan decomposition:

$$\boldsymbol{H} = \begin{bmatrix} \boldsymbol{B} & \boldsymbol{C} \\ \boldsymbol{0} & -\boldsymbol{B} \end{bmatrix} = \begin{bmatrix} \boldsymbol{I} & \boldsymbol{X} \\ \boldsymbol{0} & \boldsymbol{I} \end{bmatrix} \begin{bmatrix} \boldsymbol{B} & \boldsymbol{0} \\ \boldsymbol{0} & -\boldsymbol{B} \end{bmatrix} \begin{bmatrix} \boldsymbol{I} & \boldsymbol{X} \\ \boldsymbol{0} & \boldsymbol{I} \end{bmatrix}^{-1} \tag{18}$$

To derive the solution of Lyapunov equation, we need to utilize two important properties of matrix sign functions. Specifically, we have:

**Lemma 1.** *For a given matrix $\boldsymbol{H}$ with no eigenvalues on the imaginary axis, its sign function has the following properties: 1) $sign(\boldsymbol{H})^2 = \boldsymbol{I}$; 2) if $\boldsymbol{H}$ has the Jordan decomposition $\boldsymbol{H}=\boldsymbol{TMT}^{-1}$, then its sign function satisfies $sign(\boldsymbol{H}) = \boldsymbol{T}sign(\boldsymbol{M})\boldsymbol{T}^{-1}$.*

Lemma 1.1 shows that $sign(\boldsymbol{H})$ is the matrix square root of the identity matrix, which indicates the possibility of using Newton's root-finding method to derive the solution (Higham, 2008). Here we also adopt the Newton-Schulz iteration, the modified inverse-free and multiplication-rich Newton iteration, to iteratively compute $sign(\boldsymbol{H})$ as:

$$\begin{aligned}\boldsymbol{H}_{k+1} = \frac{1}{2}\boldsymbol{H}_k(3\boldsymbol{I} - \boldsymbol{H}_k^2) &= \frac{1}{2}\left( \begin{bmatrix} 3\boldsymbol{B}_k & 3\boldsymbol{C}_k \\ \boldsymbol{0} & -3\boldsymbol{B}_k \end{bmatrix} - \begin{bmatrix} \boldsymbol{B}_k & \boldsymbol{C}_k \\ \boldsymbol{0} & -\boldsymbol{B}_k \end{bmatrix} \begin{bmatrix} \boldsymbol{B}_k^2 & \boldsymbol{B}_k\boldsymbol{C}_k - \boldsymbol{C}_k\boldsymbol{B}_k \\ \boldsymbol{0} & \boldsymbol{B}_k^2 \end{bmatrix}\right) \\ &= \frac{1}{2}\left( \begin{bmatrix} \boldsymbol{B}_k(3\boldsymbol{I} - \boldsymbol{B}_k^2) & 3\boldsymbol{C}_k - \boldsymbol{B}_k(\boldsymbol{B}_k\boldsymbol{C}_k - \boldsymbol{C}_k\boldsymbol{B}_k) - \boldsymbol{C}_k\boldsymbol{B}_k^2 \\ \boldsymbol{0} & -\boldsymbol{B}_k(3\boldsymbol{I} - \boldsymbol{B}_k^2) \end{bmatrix}\right) \end{aligned} \tag{19}$$

The equation above defines two coupled iterations for solving the Lyapunov equation. Since the NS iteration converges only locally, *i.e.,* converges when $||\boldsymbol{H}_k^2-\boldsymbol{I}||<1$, here we divide $\boldsymbol{H}_0$ by $||\boldsymbol{B}||_{\mathrm{F}}$ to meet the convergence condition. This normalization defines the initialization $\boldsymbol{B}_0=\frac{\boldsymbol{B}}{||\boldsymbol{B}||_{\mathrm{F}}}$ and $\boldsymbol{C}_0=\frac{\boldsymbol{C}}{||\boldsymbol{B}||_{\mathrm{F}}}$, and we have the following iterations:

$$\begin{aligned} \boldsymbol{B}_{k+1} &= \frac{1}{2}\boldsymbol{B}_k(3\boldsymbol{I} - \boldsymbol{B}_k^2), \\ \boldsymbol{C}_{k+1} &= \frac{1}{2}\left( -\boldsymbol{B}_k^2\boldsymbol{C}_k + \boldsymbol{B}_k\boldsymbol{C}_k\boldsymbol{B}_k + \boldsymbol{C}_k(3\boldsymbol{I} - \boldsymbol{B}_k^2)\right). \end{aligned} \tag{20}$$

Relying on Lemma 1.2, the sign function of Eq. (18) can be also calculated as:

$$sign(\boldsymbol{H}) = sign\left(\begin{bmatrix} \boldsymbol{B} & \boldsymbol{C} \\ \boldsymbol{0} & -\boldsymbol{B} \end{bmatrix}\right) = \begin{bmatrix} \boldsymbol{I} & \boldsymbol{X} \\ \boldsymbol{0} & \boldsymbol{I} \end{bmatrix} sign\left(\begin{bmatrix} \boldsymbol{B} & \boldsymbol{0} \\ \boldsymbol{0} & -\boldsymbol{B} \end{bmatrix}\right) \begin{bmatrix} \boldsymbol{I} & \boldsymbol{X} \\ \boldsymbol{0} & \boldsymbol{I} \end{bmatrix}^{-1}$$

$$= \begin{bmatrix} \boldsymbol{I} & \boldsymbol{X} \\ \boldsymbol{0} & \boldsymbol{I} \end{bmatrix} \begin{bmatrix} \boldsymbol{I} & \boldsymbol{0} \\ \boldsymbol{0} & -\boldsymbol{I} \end{bmatrix} \begin{bmatrix} \boldsymbol{I} & -\boldsymbol{X} \\ \boldsymbol{0} & \boldsymbol{I} \end{bmatrix} \qquad (21)$$

$$= \begin{bmatrix} \boldsymbol{I} & 2\boldsymbol{X} \\ \boldsymbol{0} & -\boldsymbol{I} \end{bmatrix}$$

As indicated above, the iterations in Eq. (20) have the convergence:

$$\lim_{k\to\infty} \boldsymbol{B}_k = \mathbf{I}, \lim_{k\to\infty} \boldsymbol{C}_k = 2\boldsymbol{X} \qquad (22)$$

After iterating $k$ times, we can get the resultant approximate solution $\boldsymbol{X} = \frac{1}{2}\boldsymbol{C}_k$. Instead of manually choosing the iteration times, one can also set the termination criterion by checking the convergence $||\boldsymbol{B}_k - \boldsymbol{I}||_\mathrm{F} < \tau$, where $\tau$ is the pre-defined tolerance.

Table 2 compares the backward computation complexity of the iterative Lyapunov solver and the NS iteration. Our proposed Lyapunov solver spends fewer matrix multiplications and is thus more efficient than the NS iteration. Even if we iterate the Lyapunov solver more times (*e.g.,* 7 or 8), it still costs less time than the backward calculation of NS iteration that iterates 5 times.

## 4 EXPERIMENTS

In this section, we first perform a series of numerical tests to compare our proposed methods with the SVD and NS iteration. Subsequently, we validate the effectiveness of our MTP and MPA in two deep learning applications: ZCA whitening and covariance pooling. The ablation studies that illustrate the hyper-parameters are put in the Appendix.

### 4.1 NUMERICAL TESTS

To comprehensively evaluate the numerical performance, we compare the speed and error for the input of different batch sizes, matrices in various dimensions, different iteration times of the backward pass, and different polynomial degrees of the forward pass. In each of the following tests, the comparison is based on $10,000$ random covariance matrices and the matrix size is consistently $64\times64$ unless explicitly specified. The error is measured by calculating the Mean Absolute Error (MAE) of the matrix square root computed by the approximate methods (NS iteration, MTP, and MPA) and the accurate methods (SVD).

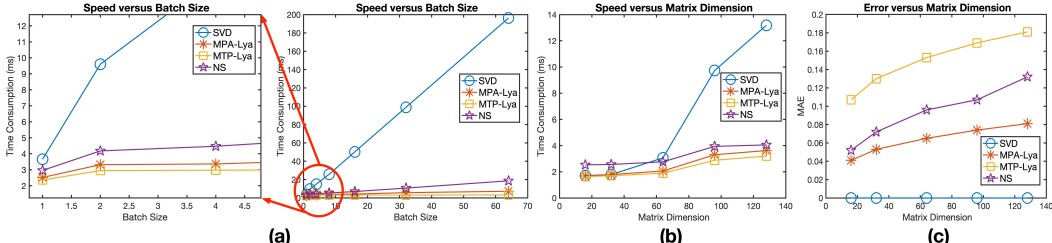

Figure 2: The results of the numerical tests. **(a)** The computation speed (FP+BP) of each method versus different batch sizes. **(b)** The speed comparison (FP+BP) of each method versus different matrix dimensions. **(c)** The error comparison of each method versus different matrix dimensions. The hyper-parameters follow our experimental setting of ZCA whitening and covariance pooling.

### 4.1.1 FORWARD ERROR VERSUS SPEED

Both the NS iteration and our methods have a hyper-parameter to tune in the forward pass, *i.e.,* iteration times for NS iteration and polynomial degrees for our MPA and MTP. To validate the

impact, we measure the speed and error for different hyper-parameters. The degrees of our MPA and MTP vary from 6 to 18, and the iteration times of NS iteration range from 3 to 7. We give the preliminary analysis in Fig. 1 (c). As can be observed, our MTP has the least computational time, and our MPA consumes slightly more time than MTP but provides a closer approximation. Moreover, the curve of our MPA consistently lies below that of the NS iteration, demonstrating our MPA is a better choice in terms of both speed and accuracy.

### 4.1.2    BACKWARD SPEED VERSUS ITERATION

Fig. 1 (d) compares the speed of our backward Lyapunov solver and the NS iteration versus different iteration times. The result is coherent with the complexity analysis in Table 2: our Lyapunov solver is much more efficient than NS iteration. For the NS iteration of 5 times, our Lyapunov solver still has an advantage even when we iterate 8 times.

### 4.1.3    SPEED VERSUS BATCH SIZE

In certain applications such as covariance pooling and instance whitening, the input could be batched matrices instead of a single matrix. To compare the speed performance for batched input, we conduct another numerical test. The hyper-parameter choices follow our experimental settings in ZCA whitening. As seen in Fig. 2 (a), our MPA-Lya and MTP-Lya are consistently more efficient than the NS iteration and SVD. To give a concrete example, when the batch size is 64, our MPA-Lya is 2.58X faster than NS iteration and 27.25X faster than SVD, while our MTP-Lya is 5.82X faster than the NS iteration and 61.32X faster than SVD.

As discussed before, the current SVD implementation adopts a for-loop to compute each matrix one by one within the mini-batch. This accounts for why the time consumption of SVD grows almost linearly with the batch size. For the NS iteration, the backward pass is not as batch-friendly as our Lyapunov solver. The gradient calculation defined in Eq. (6) requires measuring the trace and handling the multiplication for each matrix in the batch, which has to be accomplished ineluctably by a for-loop[1]. Our backward pass can be more efficiently implemented by batched matrix multiplication.

### 4.1.4    SPEED AND ERROR VERSUS MATRIX DIMENSION

In the last numerical test, we compare the speed and error for matrices in different dimensions. The hyper-parameter settings also follow our experiments of ZCA whitening. As seen from Fig. 2 (b), our proposed MPA-Lya and MTP-Lya consistently outperform others in terms of speed. In particular, when the matrix size is very small ($<32$), the NS iteration does not hold a speed advantage over the SVD. By contrast, our proposed methods still have competitive speed against the SVD. Fig. 2 (c) presents the approximation error. Our MPA-Lya always has a better approximation than the NS iteration, whereas our MTP-Lya gives a worse estimation but takes the least time consumption, which can be considered as a trade-off between speed and accuracy.

### 4.2    ZCA WHITENING: DECORRELATED BATCH NORMALIZATION

Following Wang et al. (2021), we insert the decorrelated batch normalization layer after the first convolutional layer of ResNet (He et al., 2016). For our forward pass, we match the MTP to the power series of degree 11 and set the degree for both numerator and denominator of our MPA as 5. We keep iterating 8 times for our backward Lyapunov solver. The detailed architecture changes and the implementation details of other methods are kindly referred to the Appendix.

Table 3 displays the speed and validation error on CIFAR10 and CIFAR100 (Krizhevsky, 2009). Our MTP-Lya is 1.25X faster than NS iteration and 1.48X faster than SVD-Padé, and our MPA-Lya is 1.17X and 1.34X faster. Furthermore, our MPA-Lya achieves state-of-the-art performances across datasets and models. Our MTP-Lya has comparable performances on ResNet-18 but slightly falls behind on ResNet-50. We guess this is mainly because the relatively large approximation error of MTP might affect little on the small model but can hurt the large model.

---

[1]See the code in the official Pytorch implementation of Li et al. (2018) via this link.

Table 3: Validation error of different ZCA whitening methods. The covariance matrix is of size $1 \times 64 \times 64$. The time consumption is measured for computing the matrix square root (BP+FP) on a workstation equipped with a Tesla K40 GPU and a 6-core Intel(R) Xeon(R) CPU @ 2.20GHz. For each method, we report the results based on five runs.

| Methods | Time (ms) | ResNet-18 | | | | ResNet-50 | |
| | | CIFAR10 | | CIFAR100 | | CIFAR100 | |
| | | mean±std | min | mean±std | min | mean±std | min |
|---|---|---|---|---|---|---|---|
| SVD-PI | 3.49 | 4.59±0.09 | 4.44 | 21.39±0.23 | 21.04 | 19.94±0.44 | 19.28 |
| SVD-Taylor | 3.41 | 4.50±0.08 | 4.40 | 21.14±0.20 | **20.91** | 19.81±0.24 | 19.26 |
| SVD-Padé | 3.39 | 4.65±0.11 | 4.50 | 21.41±0.15 | 21.26 | 20.25±0.23 | 19.98 |
| NS Iteration | 2.96 | 4.57±0.15 | 4.37 | 21.24±0.20 | 21.01 | **19.39±0.30** | **19.01** |
| Our MPA-Lya | 2.52 | **4.39±0.09** | **4.25** | **21.11±0.12** | 20.95 | **19.55±0.20** | 19.24 |
| Our MTP-Lya | **2.36** | 4.49±0.13 | 4.31 | 21.42±0.21 | 21.24 | 20.55±0.37 | 20.12 |

Table 4: Validation top-1/top-5 accuracy of the second-order vision transformer on ImageNet (Deng et al., 2009). The covariance matrices are of size $64 \times 48 \times 48$, where $64$ is the mini-batch size. The time cost is measured for computing the matrix square root (BP+FP) on a workstation equipped with a Tesla 4C GPU and a 6-core Intel(R) Xeon(R) CPU @ 2.20GHz. For the So-ViT-14 model, all the methods achieve similar performances but spend different epochs.

| Methods | Time (ms) | Architecture | | |
| | | So-ViT-7 | So-ViT-10 | So-ViT-14 |
|---|---|---|---|---|
| PI | **1.84** | 75.93 / 93.04 | 77.96 / 94.18 | 82.16 / 96.02 (303 epoch) |
| SVD-PI | 83.43 | 76.55 / 93.42 | 78.53 / 94.40 | 82.16 / 96.01 (278 epoch) |
| SVD-Taylor | 83.29 | 76.66 / **93.52** | 78.64 / 94.49 | 82.15 / 96.02 (271 epoch) |
| SVD-Padé | 83.25 | 76.71 / 93.49 | 78.77 / 94.51 | 82.17 / 96.02 (265 epoch) |
| NS Iteration | 10.38 | 76.50 / 93.44 | 78.50 / 94.44 | 82.16 / 96.01 (280 epoch) |
| Our MPA-Lya | 3.25 | **76.84** / 93.46 | **78.83 / 94.58** | 82.17 / 96.03 (**254** epoch) |
| Our MTP-Lya | 2.39 | 76.46 / 93.26 | 78.44 / 94.33 | 82.16 / 96.02 (279 epoch) |

### 4.3 COVARIANCE POOLING: SECOND-ORDER VISION TRANSFORMER

Now we turn to the experiment on second-order vision transformer (So-ViT). The implementation of our methods follows our ZCA whitening experiment, and other settings are put in the Appendix.

Table 4 compares the speed and performances on three So-ViT architectures with different depths. Our proposed methods predominate the SVD and NS iteration in terms of speed. To be more specific, our MPA-Lya is $3.19$X faster than the NS iteration and $25.63$X faster than SVD-Padé, and our MTP-Lya is $4.34$X faster than the NS iteration and $34.85$X faster than SVD-Padé. For the So-ViT-7 and So-ViT-10, our MPA-Lya achieves the best evaluation results and even slightly outperforms the SVD-based methods. Moreover, on the So-ViT-14 model where the performances are saturated, our method converges faster and spends fewer training epochs. The performance of our MTP-Lya is also on par with the other methods.

For the vision transformers, to accelerate the training and avoid gradient explosion, the mixed-precision techniques are often applied and the model weights are in half-precision (*i.e.,* float16). In the task of covariance pooling, the SVD often requires double precision (*i.e.,* float64) to ensure the effective numerical representation of the eigenvalues (Song et al., 2021). The SVD methods might not benefit from such a low precision, as large round-off errors are likely to be triggered. We expect the performance of SVD-based methods could be improved when using a higher precision. The PI suggested in the So-ViT only computes the dominant eigenpair but neglects the rest. In spite of the fast speed, the performance is not comparable with other methods.

## 5 CONCLUSION

In this paper, we propose two fast methods to compute the differentiable matrix square root. In the forward pass, the MTP and MPA are applied to approximate the matrix square root, while an iterative Lyapunov solver is proposed to solve the gradient function for back-propagation. Several numerical tests and computer vision applications demonstrate the effectiveness of our proposed model. In future work, we would like to extend our work to other applications of differentiable matrix square root, such as neural style transfer and covariance pooling for CNNs.

ACKNOWLEDGMENTS

This work was supported by EU H2020 SPRING No. 871245 and EU H2020 AI4Media No. 951911 projects. We thank Professor Nicholas J. Higham for valuable suggestions.

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

# A APPENDIX

## A.1 SUMMARY OF OUR ALGORITHMS

Algorithm. 1 and Algorithm. 2 summarize the forward pass (FP) and the backward pass (BP) of our proposed methods, respectively. The hyper-parameter $K$ in Algorithm. 1 means the degrees of power series, and $T$ in Algorithm. 2 denotes the iteration times.

| **Algorithm 1:** FP of our MTP and MPA. | **Algorithm 2:** BP of our Lyapunov solver. |
|---|---|
| **Input:** $\boldsymbol{A}$ and $K$ 
 **Output:** $\boldsymbol{A}^{\frac{1}{2}}$ 
 **if** *MTP* **then** 
 $\quad$ // FP method is MTP 
 $\quad \boldsymbol{A}^{\frac{1}{2}} \leftarrow \boldsymbol{I} - \sum_{k=1}^{K} \left\| \binom{\frac{1}{2}}{k} \right\| (\boldsymbol{I} - \frac{\boldsymbol{A}}{\|\boldsymbol{A}\|_{\mathrm{F}}})^k$; 
 **else** 
 $\quad$ // FP method is MPA 
 $\quad M \leftarrow \frac{K-1}{2}, N \leftarrow \frac{K-1}{2}$; 
 $\quad \boldsymbol{P}_M \leftarrow \boldsymbol{I} - \sum_{m=1}^{M} p_m (\boldsymbol{I} - \frac{\boldsymbol{A}}{\|\boldsymbol{A}\|_{\mathrm{F}}})^m$; 
 $\quad \boldsymbol{Q}_N \leftarrow \boldsymbol{I} - \sum_{n=1}^{N} q_n (\boldsymbol{I} - \frac{\boldsymbol{A}}{\|\boldsymbol{A}\|_{\mathrm{F}}})^n$; 
 $\quad \boldsymbol{A}^{\frac{1}{2}} \leftarrow \boldsymbol{Q}_N^{-1} \boldsymbol{P}_M$; 
 **end** 
 Post-compensate $\boldsymbol{A}^{\frac{1}{2}} \leftarrow \sqrt{\|\boldsymbol{A}\|_{\mathrm{F}}} \cdot \boldsymbol{A}^{\frac{1}{2}}$ | **Input:** $\frac{\partial l}{\partial \boldsymbol{A}^{\frac{1}{2}}}, \boldsymbol{A}^{\frac{1}{2}}$, and $T$ 
 **Output:** $\frac{\partial l}{\partial \boldsymbol{A}}$ 
 $\boldsymbol{B}_0 \leftarrow \boldsymbol{A}^{\frac{1}{2}}, \boldsymbol{C}_0 \leftarrow \frac{\partial l}{\partial \boldsymbol{A}^{\frac{1}{2}}}, i \leftarrow 0$; 
 Normalize $\boldsymbol{B}_0 \leftarrow \frac{\boldsymbol{B}_0}{\|\boldsymbol{B}_0\|_{\mathrm{F}}}, \boldsymbol{C}_0 \leftarrow \frac{\boldsymbol{C}_0}{\|\boldsymbol{B}_0\|_{\mathrm{F}}}$; 
 **while** $i < T$ **do** 
 $\quad$ // Coupled iteration 
 $\quad \boldsymbol{B}_{k+1} \leftarrow \frac{1}{2} \boldsymbol{B}_k (3\boldsymbol{I} - \boldsymbol{B}_k^2)$; 
 $\quad \boldsymbol{C}_{k+1} \leftarrow \frac{1}{2} \Big( -\boldsymbol{B}_k^2 \boldsymbol{C}_k + \boldsymbol{B}_k \boldsymbol{C}_k \boldsymbol{B}_k + $ 
 $\quad\quad \boldsymbol{C}_k (3\boldsymbol{I} - \boldsymbol{B}_k^2) \Big)$; 
 $\quad i \leftarrow i + 1$; 
 **end** 
 $\frac{\partial l}{\partial \boldsymbol{A}} \leftarrow \frac{1}{2} \boldsymbol{C}_k$; |

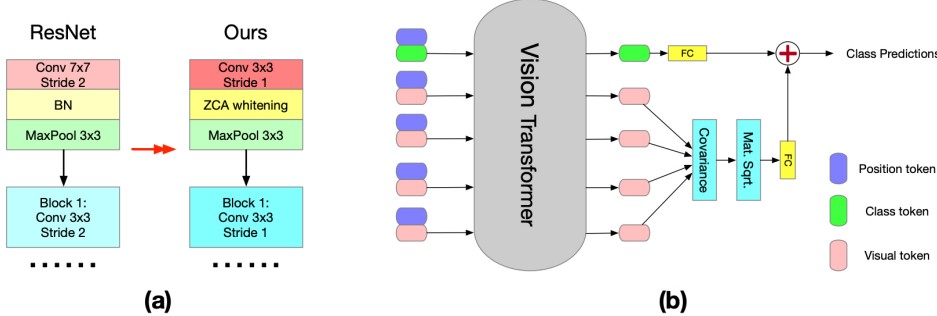

(a)        (b)

Figure 3: **(a)** The architecture changes of ResNet models in the experiment of ZCA whitening. Following Wang et al. (2021), the decorrelated batch normalization layer is inserted after the first convolutional layer. The kernel sizes, the stride of the first convolution layer, and the stride of the first ResNet block are changed correspondingly. **(b)** The scheme of So-ViT (Xie et al., 2021). The covariance square root of the visual tokens are computed to assist the classification. In the original vision transformer (Dosovitskiy et al., 2020), only the class token is utilized for class predictions.

## A.2 ZCA WHITENING

Consider the reshaped feature map $\boldsymbol{X} \in \mathbb{R}^{C \times BHW}$. The ZCA whitening procedure first computes its sample covariance as:

$$\boldsymbol{A} = (\boldsymbol{X} - \mu(\boldsymbol{X}))(\boldsymbol{X} - \mu(\boldsymbol{X}))^T + \epsilon \boldsymbol{I} \tag{23}$$

where $\boldsymbol{A} \in \mathbb{R}^{C \times C}$, $\mu(\boldsymbol{X})$ is the mean of $\boldsymbol{X}$, and $\epsilon$ is a small constant to make the covariance strictly positive definite. Afterwards, the inverse square root is calculated to whiten the feature map:

$$\boldsymbol{X}_{whitend} = \boldsymbol{A}^{-\frac{1}{2}} \boldsymbol{X} \tag{24}$$

By doing so, the eigenvalues of $\boldsymbol{X}$ are all ones, *i.e.,* the feature is uncorrelated. During the training process, the training statistics are stored for the inference phase. Compared with the ordinary batch normalization that only standardizes the data, the ZCA whitening further eliminates the correlation of the data. The detailed architecture changes of ResNet are displayed in Fig. 3 (a). For the NS iteration and our methods, we first calculate the matrix square root $\boldsymbol{A}^{\frac{1}{2}}$ and compute $\boldsymbol{X}_{whitend}$ by solving the linear system $\boldsymbol{A}^{\frac{1}{2}}\boldsymbol{X}_{whitend}=\boldsymbol{X}$.

## A.3 SECOND-ORDER VISION TRANSFORMER

For the task of image recognition, ordinary vision transformer (Dosovitskiy et al., 2020) attaches an empty class token to the sequence of visual tokens and only uses the class token for prediction, which may not exploit the rich semantics embedded in the visual tokens. Instead, So-ViT (Xie et al., 2021) proposes to leverage the high-level visual tokens to assist the task of classification:

$$y = \text{FC}(c) + \text{FC}\left((\boldsymbol{X}\boldsymbol{X}^T)^{\frac{1}{2}}\right) \tag{25}$$

where $c$ is the output class token, $\boldsymbol{X}$ denotes the visual token, and $y$ is the combined class predictions. We show the model overview in Fig. 3 (b). Equipped with the covariance pooling layer, the second-order vision transformer removes the need for pre-training on the ultra-large-scale datasets and achieves state-of-the-art performance even when trained from scratch. To reduce the computational budget, the So-ViT further proposes to use Power Iteration (PI) to approximate the dominant eigenvector and eigenvalue of the covariance $\boldsymbol{X}\boldsymbol{X}^T$.

## A.4 BASELINES

In the experiment section, we compare our proposed two methods with the following baselines:

- Power Iteration (PI). It is suggested in the original So-ViT to compute only the dominant eigenpair.
- SVD-PI (Wang et al., 2019) that uses PI to compute the gradients of SVD.
- SVD-Taylor (Wang et al., 2021; Song et al., 2021) that applies the Taylor polynomial to approximate the gradients.
- SVD-Padé (Song et al., 2021) that proposes to closely approximate the SVD gradients using Padé approximants. Notice that our MTP/MPA used in the FP is fundamentally different from the Taylor polynomial or Padé approximants used in the BP of SVD-Padé. For our method, we use Matrix Taylor Polynomial (MTP) and Matrix Padé Approximants (MPA) to derive the matrix square root in the FP. For the SVD-Padé, they use scalar Taylor polynomial and scalar Padé approximants to approximate the gradient $\frac{1}{\lambda_i-\lambda_j}$ in the BP. That is to say, their aim is to use the technique to compute the gradient and this will not involve the back-propagation of Taylor polynomial or Padé approximants.
- NS iteration (Schulz, 1933; Higham, 2008) that uses the Newton-Schulz iteration to compute the matrix square root. It has been widely applied in different tasks, including covariance pooling (Li et al., 2018) and ZCA whitening (Huang et al., 2018). We note that although Huang et al. (2019) and Higham (2008) use different forms of NS iteration, the two representations are equivalent to each other. Huang et al. (2019) just replace $\boldsymbol{Y}_k$ with $\boldsymbol{Z}_k\boldsymbol{A}$ and re-formulate the iteration using one variable. The computation complexity is still the same.

As the ordinary differentiable SVD suffers from the gradient explosion issue and easily causes the program to fail, we do not take it into account for comparison.

Unlike previous methods such as SVD and NS iteration, our MPA-Lya/MTP-Lya does not have a consistent FP and BP algorithm. However, we do not think it will bring any caveat to the stability or performance. Our ablation study in the Appendix A.7.2 implies that our BP Lyapunov solver approximates the real gradient very well (*i.e.,* $||\boldsymbol{B}_k-\boldsymbol{I}||_{\text{F}}<3e-7$ and $||0.5\boldsymbol{C}_k-\boldsymbol{X}||_{\text{F}}<7e-6$). Also, our experiments on ZCA whitening and vision transformer demonstrate superior performances. In light of these experimental results, we argue that as long as the BP algorithm is accurate enough, the inconsistency between the BP and FP is not an issue.

## A.5 IMPLEMENTATION DETAILS

All the source codes are implemented in Pytorch. For the SVD methods, the forward eigendecomposition is performed using the official Pytorch function TORCH.SVD, which calls the LAPACK's routine *gesdd* that uses the Divide-and-Conquer algorithm for the fast calculation.

All the numerical tests are conducted on a single workstation equipped with a Tesla K40 GPU and a 6-core Intel(R) Xeon(R) GPU @ 2.20GHz.

### A.5.1 TRAINING SETTINGS OF ZCA WHITENING

Suggested by Wang et al. (2020a), we truncate the Taylor polynomial to degree 20 for SVD-Taylor. To make Padé approximant match the same degree with Taylor polynomial, we set the degree of both numerator and denominator to 10 for SVD-Padé. For SVD-PI, the iteration times are also set as 20. For the NS iteration, according to the setting in Li et al. (2018); Huang et al. (2018), we set the iteration times to 5. The other experimental settings follow the implementation in Wang et al. (2021). We use the workstation equipped with a Tesla K40 GPU and a 6-core Intel(R) Xeon(R) GPU @ 2.20GHz for training.

### A.5.2 TRAINING SETTINGS OF COVARIANCE POOLING

We use 8 Tesla G40 GPUs for distributed training and the NVIDIA Apex mixed-precision trainer is used. Except that the spectral layer uses the single-precision (*i.e.,* float32), other layers use the half-precision (*i.e.,* float16) to accelerate the training. Other implementation details follow the experimental setting of the original So-ViT (Xie et al., 2021).

Following the experiment of covariance pooling for CNNs (Song et al., 2021), the degrees of Taylor polynomial are truncated to 100 for SVD-Taylor, and the degree of both the numerator and denominator of Padé approximants are set to 50 for SVD-Padé. The iteration times of SVD-PI are set to 100. In the experiment of covariance pooling, more terms of the Taylor series are used because the covariance pooling meta-layer requires more accurate gradient estimation (Song et al., 2021).

For the SVD-based methods, usually the double-precision is required to ensure an effective numerical representation of the eigenvalues. Using a lower precision would make the model fail to converge at the beginning of the training (Song et al., 2021). To resolve this issue, we first apply the NS iteration to train the network for 50 epochs, then switch to the corresponding SVD method and continue the training till the end. This hybrid approach can avoid the non-convergence of the SVD methods at the beginning of the training phase.

## A.6 EXTENSION TO INVERSE SQUARE ROOT

Although we use the LU factorization to derive the inverse square root in the paper for comparison fairness, our proposed method can naturally extend to the inverse square root. As the matrix square root $A^{\frac{1}{2}}$ of our MPA is calculated as $\sqrt{||A||_{\mathrm{F}}}Q_N^{-1}P_M$, the inverse square root can be directly computed as $\frac{1}{\sqrt{||A||_{\mathrm{F}}}}P_M^{-1}Q_N$.

## A.7 ABLATION STUDIES

We conduct three ablation studies to illustrate the impact of the degree of power series in the forward pass, the termination criterion during the back-propagation, and the possibility of combining our Lyapunov solver with the SVD and the NS iteration.

### A.7.1 DEGREE OF POWER SERIES TO MATCH FOR FORWARD PASS

Table 5 displays the performance of our MPA-Lya for different degrees of power series. As we use more terms of the power series, the approximation error gets smaller and the performance gets steady improvements from the degree $[3, 3]$ to $[5, 5]$. When the degree of our MPA is increased from $[5, 5]$ to $[6, 6]$, there are only marginal improvements. We hence set the forward degrees as $[5, 5]$ for our MPA and as 11 for our MTP as a trade-off between speed and accuracy.

Table 5: Performance of our MPA-Lya versus different degrees of power series to match.

| Degrees | Time (ms) | ResNet-18 | | | | ResNet-50 | |
| | | CIFAR10 | | CIFAR100 | | CIFAR100 | |
| | | mean±std | min | mean±std | min | mean±std | min |
|---|---|---|---|---|---|---|---|
| [3, 3] | 0.80 | 4.64±0.11 | 4.54 | 21.35±0.18 | 21.20 | 20.14±0.43 | 19.56 |
| [4, 4] | 0.86 | 4.55±0.08 | 4.51 | 21.26±0.22 | 21.03 | 19.87±0.29 | 19.64 |
| [6, 6] | 0.98 | **4.45±0.07** | 4.33 | **21.09±0.14** | 21.04 | **19.51±0.24** | 19.26 |
| [5, 5] | 0.93 | **4.39±0.09** | **4.25** | 21.11±0.12 | **20.95** | 19.55±0.20 | **19.24** |

### A.7.2 TERMINATION CRITERION FOR BACKWARD PASS

Table 6 compares the performance of backward algorithms with different termination criteria as well as the exact solution computed by the Bartels-Steward algorithm (BS algorithm) (Bartels & Stewart, 1972). Since the NS iteration has the property of quadratic convergence, the errors $||\boldsymbol{B}_k - \boldsymbol{I}||_{\mathrm{F}}$ and $||0.5\boldsymbol{C}_k - \boldsymbol{X}||_{\mathrm{F}}$ decrease at a larger rate for more iteration times. When we iterate more than 7 times, the error becomes sufficiently neglectable, *i.e.,* the NS iteration almost converges. Moreover, from 8 iterations to 9 iterations, there are no obvious performance improvements. We thus terminate the iterations after iterating 8 times.

The exact gradient calculated by the BS algorithm does not yield the best results. Instead, it only achieves the least fluctuation on ResNet-50 and other results are inferior to our iterative solver. This is because the formulation of our Lyapunov equation is based on the assumption that the accurate matrix square root is computed, but in practice we only compute the approximate one in the forward pass. In this case, calculating *the accurate gradient of the approximate matrix square root* might not necessarily work better than *the approximate gradient of the approximate matrix square root.*

Table 6: Performance of our MPA-Lya versus different iteration times. The residual errors $||\boldsymbol{B}_k - \boldsymbol{I}||$ and $||0.5\boldsymbol{C}_k - \boldsymbol{X}||_{\mathrm{F}}$ are measured based on $10,000$ randomly sampled covariance matrices.

| Methods | Time (ms) | $||\boldsymbol{B}_k - \boldsymbol{I}||_{\mathrm{F}}$ | $||0.5\boldsymbol{C}_k - \boldsymbol{X}||_{\mathrm{F}}$ | ResNet-18 | | | | ResNet-50 | |
| | | | | CIFAR10 | | CIFAR100 | | CIFAR100 | |
| | | | | mean±std | min | mean±std | min | mean±std | min |
|---|---|---|---|---|---|---|---|---|---|
| BS algorithm | 2.34 | – | – | 4.57±0.10 | 4.45 | 21.20±0.23 | 21.01 | **19.60±0.16** | 19.55 |
| #iter 5 | 1.05 | ≈0.3541 | ≈0.2049 | 4.48±0.13 | 4.31 | 21.15±0.24 | **20.84** | 20.03±0.19 | 19.78 |
| #iter 6 | 1.23 | ≈0.0410 | ≈0.0231 | 4.43±0.10 | 4.28 | 21.16±0.19 | 20.93 | 19.83±0.24 | 19.57 |
| #iter 7 | 1.43 | ≈7e−4 | ≈3.5e−4 | 4.45±0.11 | 4.29 | 21.18±0.20 | 20.95 | 19.69±0.20 | 19.38 |
| #iter 9 | 1.73 | ≈2e−7 | ≈7e−6 | **4.40±0.07** | 4.28 | **21.08±0.15** | 20.89 | **19.52±0.22** | 19.25 |
| #iter 8 | 1.59 | ≈3e−7 | ≈7e−6 | **4.39±0.09** | **4.25** | 21.11±0.12 | 20.95 | 19.55±0.20 | **19.24** |

### A.7.3 ITERATIVE LYAPUNOV SOLVER AS A GENERAL BACKWARD ALGORITHM

Notice that our proposed iterative Lyapunov solver is a general backward algorithm for computing the matrix square root. That is to say, it should be also compatible with the SVD and NS iteration as the forward pass. Table 7 compares the performance of different methods that use the Lyapunov solver as the backward algorithm. As can be seen, the SVD-Lya can achieve competitive performances compared with other differentiable SVD methods. However, the combination of Lyapunov solver with the NS iteration, *i.e.,* the NS-Lya, cannot converge on any datasets. Although the NS iteration is applied in both the FP and BP of the NS-Lya, the implementation and the usage are different. For the FP algorithm, the NS iteration is two coupled iterations that use two variables $\boldsymbol{Y}_k$ and $\boldsymbol{Z}_k$ to compute the matrix square root. For the BP algorithm, the NS iteration is defined to compute the matrix sign and only uses the variable $\boldsymbol{Y}_k$. The term $\boldsymbol{Z}_k$ is not involved in the BP and we have no control over the gradient back-propagating through it. We conjecture this might introduce some instabilities to the training process. Despite the analysis, developing an effective remedy is a direction of our future work.

### A.8 COMPUTATION ANALYSIS ON MATRIX INVERSE VERSUS SOLVING LINEAR SYSTEM

Suppose we want to compute $\boldsymbol{C} = \boldsymbol{A}^{-1}\boldsymbol{B}$. There are two options available. One is first computing the inverse of $\boldsymbol{A}$ and then calculating the matrix product $\boldsymbol{A}^{-1}\boldsymbol{B}$. The other option is to solve the

Table 7: Performance comparison of SVD-Lya and NS-Lya.

| Methods | Time (ms) | ResNet-18 | | | | ResNet-50 | |
|---|---|---|---|---|---|---|---|
| | | CIFAR10 | | CIFAR100 | | CIFAR100 | |
| | | mean±std | min | mean±std | min | mean±std | min |
| SVD-Lya | 4.47 | 4.45±0.16 | **4.20** | 21.24±0.24 | 21.02 | **19.41±0.11** | 19.26 |
| SVD-PI | 3.49 | 4.59±0.09 | 4.44 | 21.39±0.23 | 21.04 | 19.94±0.44 | 19.28 |
| SVD-Taylor | 3.41 | 4.50±0.08 | 4.40 | 21.14±0.20 | **20.91** | 19.81±0.24 | 19.26 |
| SVD-Padé | 3.39 | 4.65±0.11 | 4.50 | 21.41±0.15 | 21.26 | 20.25±0.20 | 19.98 |
| NS-Lya | 2.87 | – | – | – | – | – | – |
| NS Iteration | 2.96 | 4.57±0.15 | 4.37 | 21.24±0.20 | 21.01 | **19.39±0.30** | **19.01** |
| MPA-Lya | 2.52 | **4.39±0.09** | 4.25 | **21.11±0.12** | 20.95 | 19.55±0.20 | 19.24 |
| MTP-Lya | **2.36** | 4.49±0.13 | 4.31 | 21.42±0.21 | 21.24 | 20.55±0.37 | 20.12 |

linear system $AC=B$. This process involves first performing the GPU-friendly LU decomposition to decompose $A$ and then conducting two substitutions to obtain $C$.

Solving the linear system is often preferred over the matrix inverse for accuracy, stability, and speed reasons. When the matrix is ill-conditioned, the matrix inverse is very unstable because the eigenvalue $\lambda_i$ would become $\frac{1}{\lambda_i}$ after the inverse, which might introduce instability when $\lambda_i$ is very small. For the LU-based linear system, the solution is still stable for ill-conditioned matrices. As for the accuracy aspect, according to Higham (2008), the errors of the two computation methods are bounded by:

$$|B - AC_{inv}| \leq \alpha |A||A^{-1}||B|$$
$$|B - AC_{LU}| \leq \alpha |L||U||C_{LU}| \tag{26}$$

We roughly have the relation $|A| \approx |L||U|$. So the difference is only related to $|A^{-1}||B|$ and $|C_{LU}|$. When $A$ is ill-conditioned, $|A^{-1}|$ could be very large and the error of $|B - AC_{inv}|$ will be significantly larger than the error of linear system. About the speed, the matrix inverse option takes about $\frac{14}{3}n^3$ flops, while the linear system consumes about $\frac{2}{3}n^3 + 2n^2$ flops. Consider the fact that LU is much more GPU-friendly than the matrix inverse. The actual speed of the linear system could even be faster than the theoretical analysis.

For the above reasons, solving a linear system is a better option than matrix inverse when we need to compute $A^{-1}B$.

### A.9 SPEED COMPARISON ON LARGER MATRICES

In the paper, we only compare the speed of each method till the matrix dimension is 128. Here we add the comparison on larger matrices till the dimension 1024. We choose the maximal dimension as 1024 because it should be large enough to cover the applications in deep learning. Table 8 compares the speed of our methods against the NS iteration.

Table 8: Time consumption (ms) for a single matrix whose dimension is larger than 128. The measurement is based on $10,000$ randomly sampled matrices.

| Matrix Dimension | 1×256×256 | 1×512×512 | 1×1024×1024 |
|---|---|---|---|
| MTP-Lya | 4.88 | 5.43 | 7.32 |
| MPA-Lya | 5.56 | 7.47 | 11.28 |
| NS iteration | 6.28 | 9.86 | 12.69 |

As can be observed, our MPA-Lya is consistently faster than the NS iteration. When the matrix dimension increases, the computation budget of solving the linear system for deriving our MPA increases, but the cost of matrix multiplication also increases. Consider the huge amount of matrix multiplications of NS iteration. The computational cost of the NS iteration grows at a larger speed compared with our MPA-Lya.

Table 9: Comparison of validation accuracy on ImageNet (Deng et al., 2009) and ResNet-50 (He et al., 2016). The time consumption is measured for computing the matrix square root (FP+BP). We follow Song et al. (2021) for all the experimental settings.

| Methods | Covariance Size ($B{\times}C{\times}C$) | Time (ms) | top-1 acc (%) | top-5 acc (%) |
|---|---|---|---|---|
| MPA-Lya | $256{\times}256{\times}256$ | 110.61 | 77.13 | 93.45 |
| NS iteration | | 164.43 | 77.19 | 93.40 |

### A.10 EXTRA EXPERIMENT ON GLOBAL COVARIANCE POOLING FOR CNNS

To evaluate the performance of our MPA-Lya on batched larger matrices, we conduct another experiment on the task of Global Covariance Pooling (GCP) for CNNs (Li et al., 2017a; 2018; Song et al., 2021). In the GCP model, the matrix square root of the covariance of the final convolutional feature is fed into the FC layer for exploiting the second-order statistics. Table 9 presents the speed and performance comparison. As can be seen, the performance of our MPA-Lya is very competitive against the NS iteration, but our MPA-Lya is about $1.5$X faster.

### A.11 DEFECT OF PADÉ APPROXIMANTS

When there is the presence of spurious poles (Stahl, 1998; Baker, 2000), the Padé approximants are very likely to suffer from the well-known defects of instability. The spurious poles mean that when the approximated function has very close poles and zeros, the corresponding Padé approximants will also have close poles and zeros. Consequently, the Padé approximants will become very unstable in the region of defects (*i.e.,* when the input is in the neighborhood of poles and zeros). Generalized to the matrix case, the spurious poles can happen when the determinant of the matrix denominator is zero (*i.e.* $\det(\boldsymbol{Q}_N)=0$).

However, in our case, the approximated function is $(1-z)^{\frac{1}{2}}$ for $|z|<1$, which only has one zero at $z=1$ and does not have any poles. Therefore, the spurious pole does not exist in our approximation and there are no defects of our Padé approximants.

We can also prove this claim for the matrix case. Consider the denominator of our Padé approximants:

$$\boldsymbol{Q}_N = \boldsymbol{I} - \sum_{n=1}^{N} q_n (\boldsymbol{I} - \frac{\boldsymbol{A}}{||\boldsymbol{A}||_{\mathrm{F}}})^n \tag{27}$$

Its determinant is calculated as:

$$\det(\boldsymbol{Q}_N) = \prod_{i=1}^{N}(1 - \sum_{n=1}^{N} q_n (1 - \frac{\lambda_i}{\sqrt{\sum_i \lambda_i^2}})^n) \tag{28}$$

The coefficients $q_n$ of our $[5,5]$ Padé approximant are pre-computed as $[2.25, -1.75, 0.54675, -0.05859375, 0.0009765625]$. Let $x_i$ denotes $(1 - \frac{\lambda_i}{\sqrt{\sum_i \lambda_i^2}})$. Then $x_i$ is in the range of $[0,1]$, and we have:

$$f(x_i) = 1 - 2.25x_i + 1.75x_i^2 - 0.54675x_i^3 + 0.05859375x_i^4 - 0.0009765625x_i^5$$
$$\det(\boldsymbol{Q}_N) = \prod_{i=1}(f(x_i)) \tag{29}$$

The polynomial $f(x_i)$ does not have any zero in the range of $x{\in}[0,1]$. The minimal is $0.0108672$ when $x=1$. This implies that $\det(\boldsymbol{Q}_N)\neq 0$ always holds for any $\boldsymbol{Q}_N$ and our Padé approximants do not have any pole. Accordingly, there will be no spurious poles and defects. Hence, our MPA is deemed stable. Throughout our experiments, we do not encounter any instability issue of our MPA.

Fig. 4 depicts the function $(1-z)^{\frac{1}{2}}$ and the corresponding approximation of Taylor polynomial and Padé approximants. As can be seen, our approximating techniques do not suffer from any spurious poles or defect regions, and the stability is guaranteed. Moreover, when $z$ is close to $1$, the Taylor polynomial gets a relatively large approximation error but the Padé approximants still give a reasonable approximation. This confirms the superiority of our MPA.

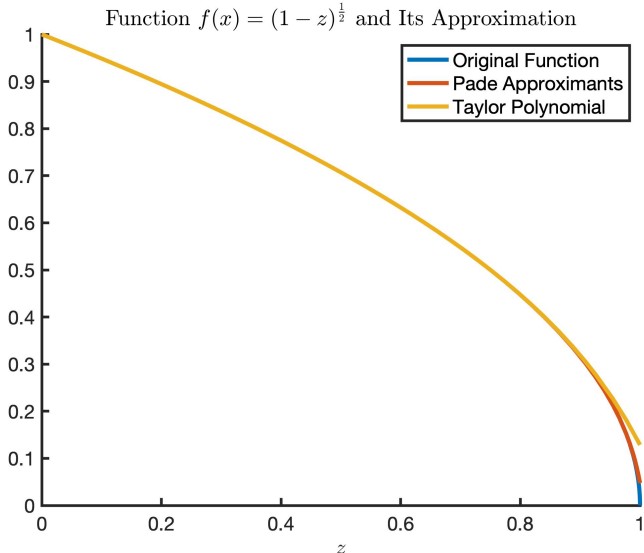

Figure 4: The function $(1 - z)^{\frac{1}{2}}$ in the range of $|z| < 1$ and its approximation including Taylor polynomial and Padé approximants. The two approximation techniques do not have any spurious poles or defect regions.

## A.12 PADÉ COEFFICIENTS OF OTHER DEGREES AND THEIR STABILITY

The property of stability also generalizes to diagonal Padé approximants of other degrees as there are no poles in the original function $(1 - z)^{\frac{1}{2}}$ for $|z| < 1$. To better illustrate this point, here we attach the Padé coefficients from the degree $[3, 3]$ to the degree $[6, 6]$. The numerator $p_m$ is:

$$p_3 = [1.75, -0.875, 0.109375].$$
$$p_4 = [2.25, -1.6875, 0.46875, -0.03515625].$$
$$p_5 = [2.75, -2.75, 1.203125, -0.21484375, 0.0107421875].$$
$$p_6 = [3.25, -4.0625, 2.4375, -0.7109375, 0.0888671875, -0.003173828125].$$

$$(30)$$

And the corresponding denominator $q_n$ is:

$$q_3 = [1.25, -0.375, 0.015625].$$
$$q_4 = [1.75, -0.9375, 0.15625, -0.00390625].$$
$$q_5 = [2.25, -1.75, 0.54675, -0.05859375, 0.0009765625].$$
$$q_6 = [2.75, -2.8125, 1.3125, -0.2734375, 0.0205078125, -0.000244140625].$$

$$(31)$$

Similar to the deduction in Eqs. (27) to (29), we can get the polynomial for deriving $\det(\boldsymbol{Q}_N)$ as:

$$f(x_i)_{q_3} = 1 - 1.25x_i + 0.375x_i^2 - 0.015625x_i^3$$
$$f(x_i)_{q_4} = 1 - 1.75x_i + 0.9375x_i^2 - 0.15625x_i^3 + 0.00390625x_i^4$$
$$f(x_i)_{q_5} = 1 - 2.25x_i + 1.75x_i^2 - 0.54675x_i^3 + 0.05859375x_i^4 - 0.0009765625x_i^5$$
$$f(x_i)_{q_6} = 1 - 2.75x_i + 2.8125x_i^2 - 1.3125x_i^3 + 0.2734375x_i^4 - 0.0205078125x_i^5 + 0.000244140625x_i^6$$

$$(32)$$

It can be easily verified that all of the polynomials monotonically decrease in the range of $x_i \in [0, 1]$ and have their minimal at $x_i = 1$ as:

$$\min_{x_i \in [0,1]} f(x_i)_{q_3} = 0.109375.$$
$$\min_{x_i \in [0,1]} f(x_i)_{q_4} = 0.03515625.$$
$$\min_{x_i \in [0,1]} f(x_i)_{q_5} = 0.0108672.$$
$$\min_{x_i \in [0,1]} f(x_i)_{q_6} = 0.003173828125.$$

$$(33)$$

As indicated above, all the polynomials are positive and we have $\det\left(\boldsymbol{Q}_N\right) \neq 0$ consistently holds for the degree $N{=}3, 4, 5, 6$.

