# OpenReview forum: "Fast Differentiable Matrix Square Root"
_ICLR.cc/2022/Conference — ICLR 2022 Poster_

### Official Review · Reviewer_Jnyr · 2021-10-24

**Correctness:** 4
**Technical Novelty And Significance:** 3
**Empirical Novelty And Significance:** 3
**Recommendation:** 8
**Confidence:** 4

**Main Review:**

I. Theoretical contributions

I.I. Forward pass: MTP and MPA approximation approaches

In terms of math, the formulae are mostly correct except for minor issues. In eq (7), the notion $(1/2 \quad k)^T$ should be further explained. In eq (11), $z^m$ is missing in the numerator and $z^n$ is missing in the denominator.

The approximation approaches were motivated by the slowness of SVD in GPU computing. We would like iterative approximations for speed. This is fair. However, while Jacobi-based SVD approaches abandon parallelism and a bit of speed to preserve stability, and orthogonality is crucial in it, the proposed approximation approaches, in my view, seem to trade stability for speed.

The proposed MTP based on Taylor expansion is fast. The authors thoroughly explained a stability issue MTP, which I find fair and honest. The explanation is used to motivate the invention of the second approach, MPA. Although it looks like MPA has addressed the stability issue in MTP with a bit of speed compromise, I think the authors have overlooked another stability issue with Pade approximants. Crucial to any Pade approximation approach is the idea that a function is approximated by the ratio of two low-degree polynomials. The problem though is that Pade approximants in scalar forms can have defects, i.e. small regions where the numerator and the denominator approaches zero. Defects can make the approximation unstable very quickly. See for example:

[1] Baker, Defects and the Convergence of Padé Approximants, In Mathematics, 2000.

One can generalise the issue when scalar serieses are replaced by matrix serieses. In eq (13), it looks like solving a linear system is key to MPA. However, what if $det(Q_N) \rightarrow 0$ and $det(P_N) \rightarrow 0$ in some region in the matrix space? This case would be like the matrix version of scalar defects.

I do not think the provided numeric tests are sufficient to spot the defects, because the defected regions can be small. Unfortunately, there is no treatment of Pade approximant defects in the paper.

I.II. Backward pass: viewing matrix square root gradient as a continuous Lyapunov equation and using the matrix sign function to solve

There is some novelty in this section but I think the novelty is not much. The idea of viewing the matrix square root gradient as a Sylvester equation is interesting but it has been proposed before (i.e. Lin-Maji (2017) pointed out in the paper). There, Lin-Maji did not use any iterative approach but instead chose to solve directly the problem using the Bartels-Stewartz (1972) algorithm. Here, the authors proposed to solve the Lyapunov equation using the matrix sign function and Newton-Schulz iteration. However, if one looks at Benner et al (2006) section 2.3 then it is clear that the proposed solution here is the same as Benner et al (2006) section 2.3 with matrices A, B, C in the Sylvester equation being replaced by $A^{1/2}$, $A^{1/2}$ and $\frac{\partial l}{\partial A^{1/2}}$.

II. Literature Review

The review of relevant work is sufficient. However, grouping together the work of Huang et al (2019) with those of Lin-Maji (2017) and Li (2018) is, in my view, not appropriate. While the latter two papers use the Newton-Schulz iteration explained in this paper, which is Higham (2008)'s approach, Huang et al (2019) has a different kind of Newton-Schulz iteration, one which deals with the matrix square root inverse rather than the matrix square root, a direct application of [2] below. Their NS iteration is more closely related to Benner et al (2006) and the one proposed in the backward pass in here. I think they deserve a separate treatment.

[2] Dario A. Bini, Nicholas J. Higham, and Beatrice Meini. Algorithms for the matrix pth root. Numerical Algorithms, 39(4):349–378, Aug 2005.

III. Experiments

The numeric tests in section 4.1 are sufficient for speed comparisons but are a bit lacking for accuracy comparisons. Fig 2c is the only test related to accuracy on general settings. While it reveals that on average, MPA is better than MTP and Higham (2008), there is no treatment about special cases where MTP or MPA may potentially fail. For example, a figure revealing when MTP fails can support the claimed "hump phenomenon" in sec 3.1.2, making a stronger point. It would be good to have some experiments on potential defects of MPA. It would be good to have some accuracy test for the backward pass as well, which is currently lacking.

Section 4.3 is adequate, showing the proposed approaches inbetween two extrema, one with full SVD (i.e. SVD-Taylor) and the other with only the principle eigenpairs (PI). However, in section 4.2 I think a better candidate to compare against is not Higham (2008) but rather Bini et al (2000) ([2] above) that Huang et al (2019) have used. They directly targeted $A^{-1/2}$ which is appropriate for ZCA whitening. If there were space, I would recommend treating the inverse matrix square root problem as a separate, but related problem. However, if the scope of the paper is just matrix square root then presenting ZCA whitening is still good but it should be stated in the paper that you are handicapped against state-of-the-art by one extra linear system.


**Summary Of The Paper:**

The paper addresses the problem of computing the matrix square root of a positive semidefinite matrix and computing the gradient of the matrix square root, so they can be used as a forward pass and a backward pass in a deep learning framework.

The paper has two separate contributions. In the forward pass, two approximation methods are introduced, Matrix Taylor Polynomial (MTP) and Matrix Pade Approximant (MPA). These are straight-forward generalisations of the scalar power series of, and the scalar Pade approximant to, the function $(1-z)^{1/2}$, to matrices.

In the backward pass, computing the gradient is viewed as a special case of the Sylvester equation, also known as the continuous Lyapunov equation, and is solved using a Newton-Schulz iteration first introduced by Benner (2006).

Numerical tests were conducted to figure the speed and error of forward and backward passes as functions of batch size and matrix dimension.

Experiments on ZCA whitening and covariance pooling in recent deep learning approaches were presented.

**Summary Of The Review:**

The paper has two separate contributions, one in forward pass and one in backward pass in dealing with (positive-semidefinite) matrix square root in deep learning.

In the forward pass, two approximation approaches MTP and MPA both seem to trade stability for speed. MTP is the fastest but also the least accurate. MPA is slightly slower but on average is more accurate. However, the current paper is not complete in the sense that it lacks treatment of "defects" (regions where the numerator and the denominator approach zero) in Pade approximant approaches. Therefore, the stability and convergence of MPA is somewhat concerning.

In the backward pass, viewing the computation of the gradient of the matrix square root as solving a Sylvester equation is interesting. However, experiments in the generic settings lack accuracy tests to reveal how close these Newton-Schulz-based approaches are to the real gradient.

There is a bit of concern when in section 4.2 the proposed approaches are not compared against the closest possible candidate (Huang et al (2019)) for the ZCA whitening problem.

---

> ### Author Response · Authors · 2021-11-12
> **Response to Reviewer Jnyr: About Stability of MPA, Accuracy of  BP solver, and Equivalence of NS iterations (Part 1/2)**
>
> We thank reviewer Jnyr for the detailed review and valuable suggestions. Our response to the questions are as follows:
>
> ---
>
> **1. Our Pad\'e approximants do not have any defects**
>
> Thanks for pointing out the concern. When there is the presence of spurious poles [1][2], the Pad\'e approximants do have the well-known defects of instability. The spurious poles mean that when the approximated function has very close poles and zeros, the corresponding Pad\'e approximants will also have close poles and zeros. Consequently, the Pad\'e approximants will become very unstable in the region of defects (_i.e.,_ when the input is in the neighborhood of poles and zeros). Generalized to the matrix case, the spurious poles can happen when the determinant of the matrix denominator is zero (_i.e.,_ $\det{(\mathbf{Q}_{N})}=0$).
>
> However, in our case, the approximated function is $(1-z)^{\frac{1}{2}}$ for $|z|<1$, which only has one zero at $z=1$ and does not have any poles. Therefore, the spurious pole does not exist in our approximation and there are no defects of our Pad\'e approximants.
>
> We can also prove this claim for the matrix case. Consider the denominator of our Pad\'e approximants:
>
> $$ \mathbf{Q_{N}}=  \mathbf{I} - \sum_{n=1}^{N} q_{n} (\mathbf{I}-\frac{\mathbf{A}}{||\mathbf{A}||_{\rm F}})^{n}$$
>
> Its determinant is calculated as:
>
> $$\det{(\mathbf{Q_{N}})}=\prod_{i=1}(1-\sum_{n=1}^{N} q_{n}(1-\frac{\lambda_{i}}{\sqrt{\sum_{i}\lambda_{i}^{2}}})^{n})$$
>
> The coefficients $q_{n}$ of our $[5,5]$ Pad\'e approximant are pre-computed as $[2.25,-1.75,0.54675,-0.05859375,0.0009765625]$. Let $x_{i}$ denotes $(1-\frac{\lambda_{i}}{\sqrt{\sum_{i}\lambda_{i}^{2}}})$. Then $x_{i}$ is in the range of $[0,1]$, and we have:
>
> $$ f( x_{i} )=1-2.25 x_{i}+1.75 x_{i}^2-0.54675 x_{i}^3+0.05859375 x_{i}^{4}-0.0009765625 x_{i}^{5},$$
>
> $$\det{(\mathbf{Q_{N}})}=\prod_{i=1}(f(x_{i}))$$
>
> The polynomial $f(x_{i})$ does not have any zero in the range of $x{\in}[0,1]$. The minimal is $0.0108672$ when $x=1$. This implies that $\det{(\mathbf{Q_{N}})}\neq0$ always holds for any $\mathbf{Q_{N}}$ and our Pad\'e approximants do not have any pole. Accordingly, there will be no spurious poles and defects. Hence, our MPA is deemed stable. Throughout our experiments, we do not encounter any instability issue of our MPA.
>
> The properties should also generalize to diagonal Pad\'e approximants of other degrees as there are no poles in the original function $(1-z)^{\frac{1}{2}}$ for $|z|<1$. We will add the above discussion in the Appendix.
>
> We will also add one figure to visualize the function $(1-z)^{\frac{1}{2}}$ and its corresponding Taylor polynomial and Pad\'e approximants. The figure could tell that our approximation technique is stable and does not have any defects. Please refer to the updated Appendix for details.
>
> >[1] Stahl, Spurious poles in Pad\'e approximation, Journal of Computational and Applied Mathematics, 1999.
> [2] Baker, Defects and the Convergence of Padé Approximants, In Mathematics, 2000.
>
> ---
>
> **2. Accuracy of backward Lyapunov solver**
>
> We measure the accuracy of our backward iterative Lyapunov solver by computing $||\mathbf{B_{k}}-\mathbf{I}||_{\rm F}$ in the ablation studies of Appendix A.7.2. Consider the convergence of our Lyapunov solver:
>
> $$\lim_{k\rightarrow\infty} \mathbf{B_{k}} = \mathbf{I},\ \lim_{k\rightarrow\infty} \mathbf{C_{k} }= 2 \mathbf{X} =2 \frac{\partial l}{\partial \mathbf{A}}$$
>
> Since the update of $\mathbf{C_{k}}$ is driven by the update of $\mathbf{B_{k}}$ and the iteration is coupled, the estimation error of our Lyapunov solver $||0.5\mathbf{C_{k}}-\mathbf{X}||$ can be approximately measured by $0.5||\mathbf{B_{k}}-\mathbf{I}||$. In Table 2 of the Appendix A.7.2, we present the error $||\mathbf{B_{k}}-\mathbf{I}||$ versus iteration times. Here we add also the error of $||0.5\mathbf{C_{k}}-\mathbf{X}||$ and display the result as follows:
>
> | Iterations      | 5 | 6 | 7 | 8 | 9 |
> | :---                                                    |    :----:   |   :----:   |   :----:   |   :----:   | :----: |
> |  $\|\|\mathbf{B_{k}}-\mathbf{I}\|\|$       | 0.3541 | 0.0410 | 7e-4 | 3e-7 | 2e-7 |
> |  $\|\|0.5\mathbf{C_{k}}-\mathbf{X}\|\|$  | 0.2049 | 0.0231 | 3.5e-4 | 7e-6 | 7e-6 |
>
> The estimation is still based on $10,000$ randomly sampled covariance matrices. As can be seen, the two errors are coherent and both reflect the convergence of the Lyapunov solver. When the iteration times are more than $7$ times, the error becomes sufficiently neglectable and we can assume our solver provides a really close approximation of the gradient.
>
> ---

---

> > ### Comment · Reviewer_Jnyr · 2021-11-13
> > **Amazing response.**
> >
> > Thank you for your detailed response. It has largely addressed my concerns in this part.
> >
> > One question regarding poles. You have shown that (scalar) $Q_5(x)$ is positive in range [0,1] and I can see from the Appendix that you have tested with $Q_3(x)$ and $Q_6(x)$. However, $Q_3(x)$ seems to reach 0 at point x around 0.9. Is that correct? Maybe can you show the first few values of $p_m$ and $q_n$ in the Appendix so that the reader can have a rough understanding of these polynomials?

---

> > > ### Author Response · Authors · 2021-11-13
> > > **Stability Generalizes to Other Degrees & Appendix Revision**
> > >
> > > Thanks for your reply and feedback. We are happy that our response could help to relieve your concern.
> > >
> > > ---
> > >
> > > We have revised the Appendix by adding the Pad\'e coefficients $p_{m}$ and $q_{n}$ from degree $[3,3]$ to degree $[6,6]$ and explaining their stabilities. The stability generalizes also to other degrees and we consistently have $\det{(\mathbf{Q_{N}})}\neq0$ for the degrees $N=3,4,5,6.$
> > >
> > > Here we attach some of the results to support our point. The denominator Pad\'e coefficient $q_{n}$ is presented below:
> > >
> > > $$q_{3}=[1.25,-0.375,0.015625].$$
> > >
> > > $$q_{4}=[1.75,-0.9375,0.15625,-0.00390625].$$
> > >
> > > $$q_{5}=[2.25,-1.75,0.54675,-0.05859375,0.0009765625].$$
> > >
> > > $$q_{6}=[2.75,-2.8125,1.3125,-0.2734375,0.0205078125,-0.000244140625].$$
> > >
> > > Notice that each set of $q_{n}$ do not have common coefficients (_we guess this is the part that might cause the misunderstanding_). Then similar with the deduction in previous response, we can get the polynomial for deriving $\det{(\mathbf{Q_{N}})}$ as:
> > >
> > > $$f(x_{i})^{q_{3}}=1-1.25x_{i}+0.375x^2_{i}-0.015625x^3_{i}$$
> > >
> > > $$f(x_{i})^{q_{4}}=1-1.75x_{i}+0.9375x^2_{i}-0.15625x^3_{i}+0.00390625x^4_{i}$$
> > >
> > > $$f(x_{i})^{q_{5}}=1-2.25x_{i}+1.75x^2_{i}-0.54675x^3_{i}+0.05859375x^4_{i}-0.0009765625x^5_{i}$$
> > >
> > > $$f(x_{i})^{q_{6}}=1-2.75x_{i}+2.8125x^2_{i}-1.3125x^3_{i}+0.2734375x^4_{i}-0.0205078125x^5_{i}+0.000244140625x^6_{i}$$
> > >
> > > It can be easily verified that all of the polynomials monotonically decrease in the range of $x_{i}{\in}[0,1]$ and have their minimal at $x_{i}{=}1$ as:
> > >
> > > $$\min_{x_{i}\in[0,1]} f(x_{i})^{q_{3}} = 0.109375.$$
> > >
> > > $$\min_{x_{i}\in[0,1]} f(x_{i})^{q_{4}} = 0.03515625.$$
> > >
> > > $$\min_{x_{i}\in[0,1]} f(x_{i})^{q_{5}} = 0.0108672.$$
> > >
> > > $$\min_{x_{i}\in[0,1]} f(x_{i})^{q_{6}} =  0.003173828125.$$
> > >
> > > As indicated above, all the polynomials are positive and $\det{(\mathbf{Q_{N}})}\neq0$ consistently holds for the degree $N{=}{3,4,5,6}$.
> > >
> > > ---

---

> > > > ### Comment · Reviewer_Jnyr · 2021-11-13
> > > > **Thank you!**
> > > >
> > > > Awesome!
> > > >
> > > > Thanks for the explanation.

---

> ### Author Response · Authors · 2021-11-12
> **Response to Reviewer Jnyr: About Stability of MPA, Accuracy of BP solver, and Equivalence of NS iterations (Part 2/2)**
>
> ---
>
> **3. NS iteration of Huang et al. (2019) is an equivalent representation of NS iteration of Higham (2008)**
>
> Thanks for pointing out the concern. We note that the NS iteration of Huang et al. (2019) derived from [3] is an equivalent representation of the NS iteration used in Higham (2008).  The NS iteration of Huang et al. (2019) uses one variable for the iteration as:
>
> $$\mathbf{Z_{k+1}}=\frac{1}{2}(3\mathbf{Z_{k}}-\mathbf{Z_{k}}^{3}\mathbf{A})$$
>
> whereas the NS iteration of Higham (2008) employs two variables to conduct the coupled iterations:
>
> $$\mathbf{Y_{k+1}}=\frac{1}{2}\mathbf{Y_{k}} (3\mathbf{I} - \mathbf{Z_{k}}\mathbf{Y_{k}}),$$
>
> $$\mathbf{Z_{k+1}}=\frac{1}{2}(3\mathbf{I}-\mathbf{Z_{k}}\mathbf{Y_{k}})\mathbf{Z_{k}}$$
>
> Now we will show how the two kinds of NS iterations are equivalent. Since $\mathbf{Z_{k}}^{-1}\mathbf{Y_{k}}=\mathbf{A}$, $\mathbf{Y_{k}}$ can be replaced with $\mathbf{Z_{k}}\mathbf{A}$ and we can re-formulate the NS iteration of Higham (2008) as:
>
> $$\mathbf{Z_{k+1}}=\frac{1}{2}(3\mathbf{Z_{k}}-\mathbf{Z_{k}}^{2}\mathbf{A}\mathbf{Z_{k}})$$
>
> Since $\mathbf{A}$ and $\mathbf{Z_{k}}$ have the same eigenspace, their matrix product commutes and we have $\mathbf{A}\mathbf{Z_{k}}=\mathbf{Z_{k}}\mathbf{A}$. Then the iteration can be further simplified as:
>
> $$\mathbf{Z_{k+1}}=\frac{1}{2}(3\mathbf{Z_{k}}-\mathbf{Z_{k}}^{3}\mathbf{A})$$
>
> As indicated above, the two kinds of NS iterations are in essence the same. The equivalence is also in the sense that both NS iterations consume the same computational complexity.
>
>
> Actually, we also thought about this question and did one experiment when drafting the manuscript. The performance comparison of the two NS iterations is presented as follows:
>
> | Methods    | Time (ms) |      ResNet-18                          |||| ResNet-50                                ||
> |-------------|-----------|------------|-------------|------------|------------|------------|------------|
> |                  |               | CIFAR10                  || CIFAR100                  ||  CIFAR100                 ||
> |                  |                |mean$\pm$std | min | mean$\pm$std | min | mean$\pm$std | min |
> | NS Iteration of Huang et al. (2019)    |2.99 | 4.53$\pm$0.14 | 4.40 | 21.28$\pm$0.17 | 21.07 | 19.53$\pm$0.35  | 19.16 |
> | NS Iteration of Higham (2008)     | 2.96 | 4.57$\pm$0.15 | 4.37 | 21.24$\pm$0.20 | 21.01 | 19.39$\pm$0.30 | 19.01 |
>
> Their performances are on par with each other on ResNet-18. But on ResNet-50, the NS iteration of Huang et al. (2019) is slightly inferior to the NS Iteration of Higham (2008). We guess the reason could be that $\mathbf{Z_{k}}^{-1}\mathbf{Y_{k}}=\mathbf{A}$ does not always strictly hold during the iterations. It holds for the initial iteration since $\mathbf{Y_{0}}=\mathbf{A}$ and $\mathbf{Z_{0}}=\mathbf{I}$. When the iteration continues, the relation might only roughly hold as $\mathbf{Z_{k}}^{-1}\mathbf{Y_{k}}\approx\mathbf{A}$. This might have a slight impact on the approximation power and the performance for large models such as ResNet-50. Thereby, we only use the NS iteration of Higham (2008) in the paper for comparison.
>
> >[3] Dario A. Bini, Nicholas J. Higham, and Beatrice Meini. Algorithms for the matrix pth root. Numerical Algorithms, 39(4):349–378, Aug 2005.
>
> ---
>
> **4. eq. (11) and notion of $(\frac{1}{2}\ k)^{T}$ in eq. (7)**
>
> The notion of $(\frac{1}{2}\ k)^{T}$ represents the binomial coefficients that involve fractions. The fraction part can be easily calculated by using the falling factorial. Thanks for pointing out the issue of eq. (11). The eq. (11) should be more formally written as:
>
> $$\frac{1-\sum_{m=1}^{M}p_{m}z^{m}}{1-\sum_{n=1}^{N}q_{n}z^{n}}  = 1 - \sum_{k=1}^{M+N} \Big|\dbinom{\frac{1}{2}}{k}\Big| z^{k}$$
>
> We will add the above revision to the paper.
>
> ---
>
> Thanks again and it is more than welcome to post any further comments or questions.

---

> > ### Comment · Reviewer_Jnyr · 2021-11-13
> > **Another amazing response!**
> >
> > Thank you for your explanation that Huang et al (2019)'s approach is equivalent to Higham (2008). Indeed, I overlooked the fact that we are dealing with symmetric positive definite matrices and the eigenspace is the same for all matrices of discussion.
> >
> > Good work!

---

> ### Comment · Reviewer_Jnyr · 2021-11-13
> **I have updated my ratings.**
>
> Since seeing the detailed responses and the updates in the revised version of the paper, I have been converted. This is a solid piece of work.
>
> I'd like to thank the authors for the paper.

---

> > ### Author Response · Authors · 2021-11-13
> > **Thanks for updating ratings!**
> >
> > Thank you for carefully reading our work and for your thoughtful comments.
> >
> > We greatly appreciate your encouraging feedback and constructive suggestions that help to shape the paper better!

---

### Official Review · Reviewer_J9af · 2021-11-01

**Correctness:** 3
**Technical Novelty And Significance:** 3
**Empirical Novelty And Significance:** 3
**Recommendation:** 8
**Confidence:** 5

**Main Review:**

Strengths:
1.	This paper starts the point of matrix multiplication times in computation of matrix square root, and proposes some strategies to reduce number of matrix multiplications both forward and backward propagations, aiming to improve the efficiency. The motivation is clear and reasonable.
2.	The idea of solving the Lyapunov equation based on matrix sign function and Newton-Schulz iteration seems interesting, which avoids huge memory consumption of the Kronecker product in the original closed-form solution and requirement of Schulz decomposition in Bartels-Stewart algorithm.
3.	The experiments are conducted on two real-world applications, including decorrelated batch normalization and second-order vision transformer. The speed-up over Newton Schultz iteration is ok.
4.	The paper is well-written and easy to read.

Weaknesses:
1.	In this paper, Matrix Taylor Polynomial or Matrix Pade Approximation are used for forward propagations, while approximate Lyapunov equation is used for backward propagation. Although Table 1 and Table 2 show MTP/MPA and Lya require less matrix multiplication than NS iteration, which one is most important for fast matrix square root? To verify it, MTP/MPA+NS based BP and NS baed FP + Lya are suggested to be compared in terms of accuracy and running time.
2.	For previous SVD-based and NS-based methods, computation processes for forward and backward propagations are consistent. However, this work adopts Matrix Taylor Polynomial or Matrix Pade Approximation for forward propagations and uses approximate Lyapunov equation for backward propagation, leading variance in forward and backward propagations. The authors would better make some discussions about this issue.
3.	The authors claimed BP of MPA is both time and memory-consuming. [r1] tries to respectively use SVD and MTP/MPA as forward and backward propagations, where the authors show BP of MPA is efficient (as shown in Table 6). The authors would better make some discussions about it.

[r1] Why Approximate Matrix Square Root Outperforms Accurate SVD in Global Covariance Pooling? ICCV, 2021.

Other comments:
1.	It is clear that MPA involves matrix inverse, which is very GPU-unfriendly.  As stated in the paper: "Moreover, we note that the matrix inverse can be avoided, as Eq. (13) can be more efficiently and numerically stably computed by solving the linear system". The authors would better provide more detailed computation and analysis.
2.	How to compute the coefficients of $p_{m}$ and $q_{n}$ for the Matrix Pade Approximation in equation (12)? Do forward operations of MPA in Table 1 contain computation of coefficients $p_{m}$ and $q_{n}$?
3.	Does Equation (2) lack a (·)_{sym} operation for $/frac{l}{U}$?
4.	Is equation (11) missing a term z^{k} in the left side?
5.	I am not sure why sign(B) in equation (21) can be calculated as identity matrix?
6.	P_{M} and Q_{N} are used to approximate the Taylor series. If I am not misunderstanding, does I- Q_{N}^{-1}P_{M} replace Q_{N}^{-1}P_{M} in Eqn. (13)? and do the terms I-X replace X in Eqn. (12)?


**Summary Of The Paper:**

This paper aims to solve an important problem: how to efficiently compute matrix square root in both forward and backward propagations. To this end, the authors proposed to exploit Matrix Taylor Polynomial and Matrix Pade Approximation to compute matrix square root in forward propagations, while approximate Lyapunov equation based on Newton Schultz iteration is developed for backward propagation. From the perspective of computation complexity, the proposed method requires less matrix multiplications than one based on Newton Schultz iteration. Experimental results on the decorrelated batch normalization and second-order vision transformer show the proposed methods are faster than existing methods, including SVD based ones and NS based ones.

**Summary Of The Review:**

This paper proposes some strategies to reduce number of matrix multiplications both forward and backward propagations, aiming to improve the efficiency of computation of matrix square root. Particularly, the idea of approximate Lyapunov equation based on matrix sign function and Newton-Schulz iteration seems interesting. The more experimental comparisons and discussions could be strengthened. Overall, I like this work.

---

> ### Author Response · Authors · 2021-11-12
> **Response to Reviewer J9af: Additional Details and Further Clarification (Part 1/3)**
>
> We thank reviewer J9af for the detailed review and valuable suggestions. Our response to the questions are as follows:
>
> ---
>
> **1. The FP/BP speed of each method**
>
> For the task of ZCA whitening, we present the detailed time consumption (ms) of the FP/BP for each method below:
>
> | Methods      | NS iteration | MPA-Lya     | MTP-Lya  |
> | :---        |    :----:   |          :----: |  :----: |
> | FP      | 1.28       | 0.93   | 0.77 |
> | BP   | 1.68        | 1.59     | 1.59 |
> | FP+BP   | 2.96       | 2.52     | 2.36 |
>
> The time is measured on the $1{\times}64{\times}64$ matrix. In the ZCA whitening task, the speed advantage mainly comes from our MPA and MTP. However, this does not mean that our Lyapunov solver always has a marginal advantage over the NS iteration. When the input scales to batched matrices, the speed of our Lyapunov solver will become much more advantageous over the BP of NS iteration. Consider the time consumption of FP/BP in the second-order vision transformer where the input is a mini-batch of matrices of size $64{\times}48{\times}48$:
>
> | Methods      | NS iteration | MPA-Lya     | MTP-Lya  |
> | :---        |    :----:   |          :----: |  :----: |
> | FP      | 1.75 | 1.62 | 0.78 |
> | BP   | 8.63 | 1.63 | 1.61 |
> | FP+BP   | 10.38 | 3.25 | 2.39 |
>
> As we can see, for batched matrices, a large percentage of the speed advantage comes from the BP of our Lyapunov solver.
>
> ---
>
> **2. Compatibility of MTP/MPA-NS and NS-Lya**
>
> Unfortunately, the NS-based back-propagation method is not compatible with our MPA/MTP solver. That is because the back-propagation of NS iteration requires all the intermediate variables $Y_{0}$, $Y_{1}$, $\dots$, $Y_{k}$ and $Z_{0}$, $Z_{1}$, $\dots$, $Z_{k}$ during the iterations. However, our MPA/MTP only has the final approximation of the matrix square root. Therefore, our MPA/MTP as the forward propagation is not compatible with the NS-based backward method.
>
> Nonetheless, our Lyapunov solver can work as a general backward algorithm for any method that computes the matrix square root, including SVD and NS iteration. We illustrate this in the ablation studies of Appendix A.7.3. Here we attach the result and analysis.
>
> | Methods    | Time (ms) |      ResNet-18              |||| ResNet-50                ||
> |-------------|-----------|------------|-------------|------------|------------|------------|------------|
> |                  |               | CIFAR10                  || CIFAR100                  ||  CIFAR100                 ||
> |                  |                |mean$\pm$std | min | mean$\pm$std | min | mean$\pm$std | min |
> |SVD-Lya    |4.47         |4.45$\pm$0.16 |4.20 | 21.24$\pm$0.24 | 21.02 | 19.41$\pm$0.11 |19.26|
> |NS-Lya      |2.87        |  -- |-- | -- | -- | -- | -- |
>
> As can be seen, the SVD-Lya can achieve competitive performances compared with other differentiable SVD methods ( _e.g.,_ SVD-Taylor or SVD-Pad\'e). However, the NS-Lya, cannot converge on any datasets. Although the NS iteration is applied in both the FP and BP of the NS-Lya, the implementation and the usage are different. For the FP algorithm, the NS iteration is two coupled iterations that use two variables $Y_{k}$ and $Z_{k}$ to compute the matrix square root. For the BP algorithm, the NS iteration is defined to compute the matrix sign function and only uses the variable $Y_{k}$. The term $Z_{k}$ is not involved in the BP and we have no control over the gradient back-propagating through it. We conjecture this might introduce some instabilities to the training process.
>
> ---
>
> **3. Forward and backward consistency of our methods**
>
> Unlike previous methods such as SVD and NS iteration, our MPA-Lya/MTP-Lya does not have a consistent FP and BP algorithm. However, we do not think it will bring any caveat to the stability or the performance. Our ablation studies in the Appendix A.7.2 implies that our BP Lyapunov solver approximates the real gradient very well ( _i.e.,_ $||B_{k}-I||<3e-7$ and $||0.5 C_{k}-X ||<7e-6$). Also, our experiments on ZCA whitening and vision transformer demonstrate superior performances. In light of these experimental results, we argue that as long as the BP algorithm is accurate enough, the inconsistency between the BP and FP is not an issue.
>
> ---

---

> ### Author Response · Authors · 2021-11-12
> **Response to Reviewer J9af: Additional Details and Further Clarification (Part 2/3)**
>
> ---
>
> **4. Difference with SVD-Pad\'e**
>
> Thanks for pointing out the concern. We note that our MTP/MPA used in the FP is fundamentally different from the Taylor polynomial or Pad\'e approximants used in the BP of SVD-Pad\'e [1].
>
> For our method, we use Matrix Taylor Polynomial (MTP) and Matrix Pad\'e Approximants (MPA) to derive the matrix square root in the FP. For SVD-Pad\'e, they use scalar Taylor polynomial and scalar Pad\'e approximants to approximate the gradient $\frac{1}{\lambda_{i}-\lambda_{j}}$ in the BP. That is to say, their aim is to use the technique to compute the gradient and this will not involve the back-propagation of Taylor polynomial or Pad\'e approximants. So the SVD-Pad\'e is free of the time-consuming issue of the BP.
>
> >[1] Song, Why Approximate Matrix Square Root Outperforms Accurate SVD in Global Covariance Pooling? in ICCV, 2021.
>
> ---
>
> **5. Computation analysis of matrix inverse and solving linear system**
>
> Suppose we want to compute $\mathbf{C}=\mathbf{A}^{-1}\mathbf{B}$. There are two options available. One is first computing the inverse of $\mathbf{A}$ and then calculating the matrix product $\mathbf{A}^{-1}\mathbf{B}$. The other option is to solve the linear system $\mathbf{A}\mathbf{C}=\mathbf{B}$. This process involves first performing the GPU-friendly LU decomposition to decompose $\mathbf{A}$ and then conducting two substitutions to obtain $\mathbf{C}$.
>
> Solving the linear system is often preferred to the matrix inverse for accuracy, stability, and speed reasons. When the matrix is ill-conditioned, the matrix inverse is very unstable because the eigenvalue $\lambda_{i}$ would become $\frac{1}{\lambda_{i}}$ after the inverse, which might introduce instability when $\lambda_{i}$ is very small. For the LU-based linear system, the solution is still stable for ill-conditioned matrices. As for the accuracy aspect, according to [2], the errors of the two computation methods are bounded by:
>
> $$|\mathbf{B} - \mathbf{A}\mathbf{C_{inv}|}\leq \alpha|\mathbf{A}||\mathbf{A}^{-1}||\mathbf{B}|$$
>
> $$|\mathbf{B} - \mathbf{A}\mathbf{C_{LU}}|\leq \alpha |\mathbf{L}||\mathbf{U}||\mathbf{C_{LU}}|$$
>
> We roughly have the relation $|\mathbf{A}|\approx|\mathbf{L}||\mathbf{U}|$. So the difference is only related to $|\mathbf{A}^{-1}||\mathbf{B}|$ and $|\mathbf{C_{LU}}|$. When $\mathbf{A}$ is ill-conditioned, $|\mathbf{A}^{-1}|$ could be very large and the error of $|\mathbf{B} - \mathbf{A}\mathbf{C_{inv}}|$ will be significantly larger than the error of linear system. About the speed, the matrix inverse option takes about $\frac{14}{3}n^{3}$ flops, while the linear system consumes about $\frac{2}{3}n^{3}+2n^{2}$ flops. Consider the fact that LU is much more GPU-friendly than the matrix inverse. The actual speed of the linear system could even be faster than the theoretical analysis.
>
> For the above reasons, solving a linear system is a better option than matrix inverse when we need to compute $\mathbf{A}^{-1}\mathbf{B}$.
>
> >[2] Higham, Function of Matrices, 2008
>
> ---
>
> **6. Symbol $(·)_{sym}$ with eq. (2)**
>
> Thanks for pointing out the concern. The eq. (2) does not lack the symbol $(·)_{sym}$. Throughout the paper, we do not make any distinction between symmetric and unsymmetric matrices. Also, we do not perform any matrix symmetrication.
>
> ---
>
> **7. Issue with eq. (11) and the relevant eq. (12) and eq. (13)**
>
> Thanks for pointing out the issue of eq. (11). The eq. (11) should be more formally written as:
>
> $$\frac{1-\sum_{m=1}^{M}p_{m}z^{m}}{1-\sum_{n=1}^{N}q_{n}z^{n}}  = 1 - \sum_{k=1}^{M+N} \Big|\dbinom{\frac{1}{2}}{k}\Big| z^{k}$$
>
> We will add the above revision to the paper. The eq. (12) and eq. (13) in the paper are correct. They are the generalization of Pad\'e approximants to the matrix case that corresponds to the updated eq. (11).
>
> ---
>
> **8.  Computation of Pad\'e coefficients**
>
> To compute the Pad\'e coefficients, we need to match the Pad\'e approximants to the corresponding Taylor polynomial as shown in the updated eq. (11). This matching gives rise to a system of linear equations, where each equation defines the matching at one degree:
>
> $$-\Big|\dbinom{\frac{1}{2}}{1}\Big|-q_{1} =  -p_{1}$$
>
> $$-\Big|\dbinom{\frac{1}{2}}{2}\Big|+\Big|\dbinom{\frac{1}{2}}{1}\Big|q_{1}-q_{2} =-p_{2}$$
>
> $$-\Big|\dbinom{\frac{1}{2}}{M}\Big|+\Big|\dbinom{\frac{1}{2}}{M-1}\Big|q_{1}+\dots-q_{M}=-p_{M}$$
>
> $$\dots$$
>
> Then we can compute the coefficients $p_{m}$ and $q_{n}$ by solving the above linear system.
>
> We note that the coefficients only need to be pre-computed once before inserting the meta-layer into the neural network. Then during the forward and backward propagation, the coefficients are fixed and will not be updated anymore.
>
> ---

---

> ### Author Response · Authors · 2021-11-12
> **Response to Reviewer J9af: Additional Details and Further Clarification (Part 3/3)**
>
> ---
>
> **9. $sign(\mathbf{B})$ calculated as identify matrix**
>
> The matrix sign function is a generalization of the scalar sign function. Its value depends on the positivity of the eigenvalue of the matrix (_i.e.,_ $+1$ for positive eigenvalue and $-1$ for negative eigenvalue). Since our $\mathbf{B}$ is initialized with the matrix square root $\mathbf{A}^{\frac{1}{2}}$ that is positive definite, there will not be any negative eigenvalues for $\mathbf{B}$. Therefore, the relation satisfies $sign(\mathbf{B}){=}\mathbf{I}$.
>
> ---
>
> Thanks again and it is more than welcome to post any further comments or questions.

---

### Official Review · Reviewer_JTDk · 2021-11-03

**Correctness:** 4
**Technical Novelty And Significance:** 2
**Empirical Novelty And Significance:** 3
**Recommendation:** 8
**Confidence:** 4

**Main Review:**

Strength:

+ The proposed method is differentiable and can be applied to a variety of deep learning tasks.

+ MTA and MPA tradeoff between speed and accuracy, which provide users flexible options.

+ The writing is clear and easy to follow.


Weaknesses and questions:

- The novelty of the proposed method is limited and the theory behind it is well-known.

- The speedup by the proposed method is not very significant compared to NS iteration for large matrix dimension, see Figure 2(b).
MPA requires either matrix inverse or solving a linear system of equations, which should be more computationally expensive than matrix multiplication. This may explain that when increasing matrix dimension, in Figure 2(b) the runtime of MPA becomes closer to NS iteration. My question is that when further increasing the matrix dimension (>200), will MPA-Lya be slower than NS?

- Most of the experiments are on matrices of small dimensions. I wonder what the maximal dimension the proposed method may handle (probably comparable to that of NS)? Of course it should also depend on the space complexity.


**Summary Of The Paper:**

The work proposes a fast method to solve the matrix square root. The proposed approach is differentiable and is faster than SVD and NS iteration. Thus, the proposed method is very suitable for optimizing deep neural networks that involves matrix square root computation. Its forward pass uses matrix taylor polynomial or matrix pade approximants. Its backward propagation is done by solving continuous-time Lyapunov equation. Various experiments demonstrate its better performance in both accuracy and speed.


**Summary Of The Review:**

In summary, I believe that the proposed method can be useful for a variety of deep learning tasks, although clarification of scalability of the proposed method on large matrices is expected.

---

> ### Author Response · Authors · 2021-11-12
> **Response to Reviewer JTDk: Speed Comparison and New Experiment on Larger Matrices**
>
> We thank reviewer JTDk for the detailed review and valuable suggestions. Our response to the questions are as follows:
>
> ---
>
> **1. Our MPA-Lya is still faster than the NS iteration on larger matrices**
>
> In the paper, we compare the speed of each method till the matrix dimension is $128$. It might seem that the margin between our MPA-Lya and NS iteration is narrowing as the dimension increases, and at a certain point our method would no longer have an advantage over the NS iteration. However, when the matrix dimension increases, the computation budget of solving the linear system for deriving our MPA increases, but the cost of matrix multiplication also increases. Consider the huge amount of matrix multiplications of NS iteration. The computational cost of NS iteration grows at a larger speed compared with our MPA-Lya. The time consumption (ms) for a single matrix of dimensions larger than $128$ is presented below:
>
> | Matrix Dimension      | $1{\times}256{\times}256$ | $1{\times}512{\times}512$     | $1{\times}1024{\times}1024$     |
> | :---        |    :----:   |    :----:   |          :----: |
> | MTP-Lya     | 4.88       | 5.43   |  7.32   |
> | MPA-Lya     | 5.56       | 7.47   |  11.28   |
> | NS iteration   | 6.28       | 9.86     | 12.69     |
>
> The evaluation is based on $10,000$ randomly sampled matrices. We compare the speed until the matrix dimension is $1024{\times}1024$, which should be large enough to cover the applications in deep learning. As can be observed, our MPA-Lya is consistently faster than the NS iteration. We will add the above analysis in the Appendix.
>
> ---
>
> **2. Experiment on Global Covariance Pooling for CNNs**
>
> As discussed in the paper, since our method does not need to handle the back-propagation of matrix trace, our MPA-Lya is more batch-friendly than the NS iteration. To evaluate the performance of our MPA-Lya on batched larger matrices (${>}200$), we conduct another experiment on the task of Global Covariance Pooling (GCP) for CNNs [1]. In the GCP model, the matrix square root of the covariance of the final convolutional feature is fed into the FC layer for exploiting the second-order statistics. We display the speed and performance comparison on ImageNet based on ResNet-50 as follows:
>
> |  Methods      | Covariance Size ($B{\times}C{\times}C$) | Time (ms)     | top-1 acc (\%)     | top-5 acc (\%)     |
> | :---        |    :----:   |    :----:   |     :----:   |          :----: |
> | MPA-Lya    | $256{\times}256{\times}256$      | 110.61  |  77.13   | 93.45   |
> | NS iteration    |  $256{\times}256{\times}256$      | 164.43  |  77.19   | 93.40   |
>
> The time consumption is measured on a single step (FP+BP). As can be seen, the performance of our MPA-Lya is very competitive against the NS iteration, but our MPA-Lya is about $1.5$X faster. We will also add this experiment in the Appendix.
>
> > [1] Li, Towards Faster Training of Global Covariance Pooling Networks by Iterative Matrix Square Root Normalization, in CVPR, 2018
>
>
> ---
>
> Thanks again and it is more than welcome to post any further comments or questions.

---

### Author Response · Authors · 2021-11-12
**To All Reviewers: Updated Manuscript and Appendix**

We thank all the reviewers for their constructive feedback and comments. Besides the individual responses, we have revised our paper by taking into account some suggestions. The main changes are summarized below:

---



1) @$\textcolor{red}{Reviewer\ JTDk}$: Add the speed comparison on larger matrices till the dimension $1024$ to the Appendix;

2) @$\textcolor{red}{Reviewer\ JTDk}$: Add the experiment of global covariance pooling for CNNs where the matrix size is larger than $200$ to the Appendix.

3) @$\textcolor{blue}{Reviewer\ J9af}$: Clarify the difference of our method with SVD-Pad\'e in Appendix A.4.

4) @$\textcolor{blue}{Reviewer\ J9af}$: Discuss why the BP/FP inconsistency has no impact on our methods in Appendix A.4.

5) @$\textcolor{blue}{Reviewer\ J9af}$: Add the computation analysis of matrix inverse versus solving the linear system to the Appendix.

6) @$\textcolor{blue}{Reviewer\ J9af}$ and $\textcolor{green}{Reviewer Jnyr}$: Update some math notions and equations in the paper, including eq. (11) and the notion in eq. (6).

7) @$\textcolor{green}{Reviewer\ Jnyr}$: Explain why our MPA is consistently stable and there are no defect regions in the Appendix, including one short proof and one visual illustration.

8) @$\textcolor{green}{Reviewer\ Jnyr}$: Add the Pad\'e coefficients $p_{m}$ and $q_{n}$ from degree $[3,3]$ to $[6,6]$ and explain that the stability also generalizes to other degrees in the Appendix.

9) @$\textcolor{green}{Reviewer\ Jnyr}$: Update the estimation error of backward gradient in the Appendix A.7.2 by measuring also $||0.5\mathbf{C_{k}}-\mathbf{X}||_{\rm F}$.

10) @$\textcolor{green}{Reviewer\ Jnyr}$: Clarify the equivalence of two NS iterations of Huang et al. (2019) and Higham (2008) in Appendix A.4.

---

---

### Decision · Program_Chairs · 2022-01-20

**Decision:**

Accept (Poster)

**Comment:**

The paper presents some efficiency improvements over existing methods to compute matrix square root and its gradient.  Reviewers find that the novelty over existing methods is sufficient, and that the improvements are valuable.

I propose a poster despite the relatively high numerical scores, because the group of practitioners who will use the result is somewhat niche -- the reviewers are of course selected from this group and hence value the paper more highly.

In addition the real-world speedups are modest, but it is nevertheless important to document this approach.